

# Impact of formulations of the nucleation rate on ice nucleation events

Peter Spichtinger[1], Patrik Marschalik[1,2], and Manuel Baumgartner[1,3,4]

[1] Institute for Atmospheric Physics, Johannes Gutenberg University Mainz, Germany
[2] now at DB Systel GmbH, Berlin, Germany
[3] Zentrum für Datenverarbeitung, Johannes Gutenberg University Mainz, Germany
[4] now at German Weather Service (DWD), Offenbach, Germany

**Correspondence:** Peter Spichtinger (peter.spichtinger@uni-mainz.de)

**Abstract.** Ice formation in cold temperature regimes is most probably dominated by homogeneous freezing of aqueous solution droplets. The nucleation rate as derived from laboratory experiments can be represented as a function of water activity. For idealized nucleation events as modelled with a state-of-the-art ice microphysics, the impact of different approximations of the nucleation rate on the resulting ice crystal number concentrations and maximum supersaturation ratios is investigated. The nucleation events are sensitive to the slope of the nucleation rate but only weakly affected by changes in its absolute value. This leads to the conclusion that details of the nucleation rate are less important for simulating ice nucleation in bulk models, if the main feature of the nucleation rate (i.e. its slope) is represented sufficiently. The weak sensitivity on the absolute values of the nucleation rate suggests that the amount of available solution droplets also does not crucially affect nucleation events. The use of just one distinct nucleation threshold for analysis and model parameterisation should be reinvestigated. The frequently used thresholds corresponding to a very high nucleation rate value is not reached in many nucleation events with low vertical updrafts. In contrast, the maximum supersaturation and thus the nucleation thresholds reached during an ice nucleation event depend on the vertical updraft velocity or cooling rate. This feature might explain some high supersaturation values during nucleation events in cloud chambers and suggests a reformulation of ice nucleation schemes used in coarse models based on a fixed nucleation threshold.

## 1 Introduction and theory of nucleation

Clouds are one of the most important components in the Earth-Atmosphere system. They influence the hydrological cycle and Earth's energy balance via interaction with radiation. Clouds can cool the system by partly scattering and reflecting incoming solar radiation (albedo effect) but also warm the atmosphere by absorbing and re-emitting thermal radiation as emitted by the Earth's surface (greenhouse effect). While for liquid clouds a net cooling effect can be derived, the radiative effect for clouds containing ice crystals is still under debate. In particular for pure ice clouds (so-called cirrus clouds) at high altitudes in the low temperature range ($T < 235\,\mathrm{K}$) albedo effect and greenhouse effect are of the same order of magnitude but admit different signs, leading to different net-effects (see, e.g., Fusina et al., 2007; Joos et al., 2014; Gasparini et al., 2017). Thus, details in microphysical properties of ice crystals might decide about a net warming or cooling of cirrus clouds, as can be seen in





former model studies (e.g. Zhang et al., 1999). A key aspect of ice crystals is their size which directly affects the scattering

and absorption of radiation. Smaller crystals scatter incoming solar light more effectively, thus the optical depth $\tau$ is directly

dependent on the size, as can be seen in the usual approximation (cf., e.g., Fu and Liou, 1993)

$$\tau = \text{IWC} \cdot \Delta z \cdot \left( a + \frac{b}{D_e} \right), \tag{1}$$

where $D_e$ denotes the effective diameter of the crystal, IWC is the ice water content, and $\Delta z$ represents the vertical extent of

the cloud, respectively; $a$ and $b$ are empirically derived constants. Since the available water vapour is mainly determined by

thermodynamic conditions, the pathway of ice nucleation often decides about the ice crystal number concentration in cirrus

clouds and thus their effective size (assuming a certain amount of available water vapour).

Ice crystals can be formed by very different nucleation processes, which can be grouped into two major pathways, namely *in

situ* and *liquid origin* ice formation (e.g. Krämer et al., 2016; Luebke et al., 2016; Wernli et al., 2016). The overall term *in situ*

*formation* refers to ice formation at humidities below water saturation, whereas *liquid origin formation* subsumes all formation

processes where cloud droplets are present and humidity is close to water saturation (e.g. freezing of cloud droplets). It is well

known, that the ice crystal number concentration varies crucially in dependence on the underlying nucleation process, leading

to potentially strong changes in the resulting radiative effect (see, e.g., Krämer et al., 2020).

Despite of the availability of many observational data and laboratory experiments (e.g. Hoose and Möhler, 2012), and also the

development of new theoretical models (e.g. the soccer ball model, see Niedermeier et al., 2011), the details of ice nucleation

at the molecular scale are still unknown.

A special situation occurs for the probably dominant formation process at cold temperatures below $235\,\text{K}$, the so-called

homogeneous freezing of super-cooled solution droplets (also short: homogeneous nucleation). This process describes the

spontaneous freezing of supercooled aqueous solution particles containing a small amount of (usually inorganic) substances.

Albeit also the details of this freezing process are not completely understood on a molecular scale, reproducible laboratory

experiments allowed the formulation of an empirical fit for the nucleation rate (Koop et al., 2000).

In the following, the nucleation events are investigated in the phase space spanned by temperature and water activity of the

aqueous solution. The latter is defined as the ratio of saturation pressures of water vapour over the solution $p_{\text{sol}}$ and pure water

$p_{\text{liq}}$, respectively, as $a_w := \frac{p_{\text{sol}}}{p_{\text{liq}}}$. In this representation, the melting curve for different inorganic solutions turns out to be solely

temperature dependent, i.e. $a_w^i(T) := a_w(T_m) = \frac{p_{\text{ice}}(T)}{p_{\text{liq}}(T)}$ (cf. Koop, 2015, his eq.(5)), where $p_{\text{ice}}$ denotes the saturation vapour

pressure over ice. The important insight here is that also the freezing/nucleation events collapse to a single line in the diagram

(see Koop et al., 2000; Koop, 2004, 2015), which can be fitted by shifting the melting curve (deviation $\Delta a_w \sim 0.305$). This

also means that the nucleation events do not depend on the solute, which is at least true for most inorganic substances (see, e.g.,

Koop, 2004). Thus, the nucleation rate can be solely parameterized as a function of $\Delta a_w = a_w - a_w^i$. For the fitting procedure

in Koop et al. (2000), a polynomial of degree 3 is used and results in the formulation

$$J_{\text{sol}}(\Delta a_w) = 10^{p_3(\Delta a_w)} \quad \text{with} \quad p_3(x) = \sum_{k=0}^{3} a_k x^k \tag{2}$$





of the homogeneous nucleation rate coefficient $J_{\mathrm{sol}}$. The nucleation rate coefficient is used to formulate the probability of freezing of aqueous solution droplets. The fit was used in the spirit of the representation of the nucleation rate for pure water as derived by Pruppacher (1995). During this time, three water theories were available, and the nucleation rate was chosen according to the stability limit hypothesis (e.g. Mishima and Stanley, 1998). However, meanwhile this water theory can be

ruled out by experimental evidence, thus only the two other water theories remain (singularity-free hypothesis vs. liquid-liquid critical point, cf. Gallo et al., 2019, 2016). For homogeneous freezing of solution droplets at atmospheric relevant conditions, both theories produce essentially the same results. Only at very low temperatures $T < 150\,\mathrm{K}$, where highly viscous or even glassy states of water occur, a different behaviour is predicted. However, these temperatures are not relevant for investigations of ice clouds in the tropopause region, where homogeneous freezing of solution droplets appears as the dominant freezing

process. Finally, using the assumption of solution droplets being in equilibrium with their environment, water activity equals the liquid water saturation ratio $S_{\mathrm{liq}}$ due to

$$a_w = \frac{p_{\mathrm{sol}}}{p_{\mathrm{liq}}} \overset{\text{in eq.}}{=} \frac{p_v}{p_{\mathrm{liq}}} = S_{\mathrm{liq}} \tag{3}$$

where $p_v$ denotes the partial water vapour pressure. Using this representation of $a_w$ together with the ice saturation ratio $S_i = \frac{p_v}{p_{\mathrm{ice}}}$, the computation

$$
\begin{aligned}
\Delta a_w = a_w - a_w^i &= \frac{p_v}{p_{\mathrm{liq}}(T)} - \frac{p_{\mathrm{ice}}(T)}{p_{\mathrm{liq}}(T)} = (S_i - 1)\frac{p_{\mathrm{ice}}(T)}{p_{\mathrm{liq}}(T)} \\
&= (S_i - 1)a_w^i(T)
\end{aligned}
\tag{4}
$$


shows that $\Delta a_w$ only depends on the ice saturation ratio and temperature.

From theory (e.g. Baumgartner and Spichtinger, 2019) and former idealized box model simulations (e.g. Kärcher and Lohmann, 2002; Ren and Mackenzie, 2005; Spichtinger and Gierens, 2009), we know that ice crystal numbers as produced in nucleation events driven by a constant cooling rate (equivalent to a constant vertical velocity) crucially depend on several

parameters and, thus, affect also the radiative properties of the formed ice cloud (see, e.g., calculations in Krämer et al., 2020; Joos et al., 2009). In this study we will investigate the impact of the formulation of the nucleation rate on the resulting ice crystal number concentrations. In an upcoming study, the impact of ice crystal growth on nucleation events is investigated. Both processes (nucleation and growth) determine the nucleation events directly. However, the representation of these processes contains still uncertain parameters or even the (mathematical) formulation of the processes remain uncertain. Therefore, it is

of high interest to investigate the dependence of the ice nucleation events on these processes and their formulations.

The study is structured as follows. In the next section, we present theoretical concepts and the simple model used for idealized simulations for testing the impact of different processes. In section 3 the formulation of the nucleation rate along with several approximations is discussed. The consequences of using the proposed approximations are explored by idealized numerical simulations. In section 4 we investigate the impact of a recently proposed formulation of the saturation vapour pressure over

super-cooled liquid water on the nucleation events (Nachbar et al., 2019). In section 5 a new formulation of the nucleation rate based on results for freezing of pure super-cooled water (Koop and Murray, 2016) is presented and its impact on the number



concentration of nucleated ice crystals is discussed. In section 6 we investigate thresholds of ice nucleation as well as the peak values of supersaturation during nucleation events, Finally, we summarise these results and draw some conclusions in section 7.

## 2 Model description

We begin with the description of the governing equations for the relevant ice processes, i.e. homogeneous nucleation and diffusional growth. Both processes are key for determining the properties of the nucleation event, such as the number of nucleated ice crystals and the evolution of the ice saturation ratio (e.g. peak value). Of course, other processes such as sedimentation and aggregation of ice crystals are important for the evolution of ice clouds, but usually act on larger time scales, e.g., when the particles are grown to larger sizes. Thus, we omit these processes and concentrate on nucleation and growth, as in former studies (e.g. Kärcher and Lohmann, 2002; Baumgartner and Spichtinger, 2019).

We formulate the model in terms of averaged quantities for ice crystal mass and number concentration $(q_i, n_i)$. Additionally, the saturation ratio with respect to hexagonal ice, $S_i = \frac{p_v}{p_{\text{ice}}(T)}$, is used, with the partial water vapour pressure $p_v$ and the saturation water vapour pressure over hexagonal ice, $p_{\text{ice}}(T)$, respectively. Thus, the complete set of equations for an adiabatically ascending air parcel can be represented as follows

$$\dot{n}_i = \text{Nuc}_n \tag{5}$$

$$\dot{q}_i = \text{Nuc}_q + \text{Dep}_q \tag{6}$$

$$\dot{S}_i = \text{Cool} + \text{Dep}_s \tag{7}$$

$$\dot{T} = \left.\frac{\mathrm{d}T}{\mathrm{d}t}\right|_{\text{adiabatic}} + \left.\frac{\mathrm{d}T}{\mathrm{d}t}\right|_{\text{diabatic}} \tag{8}$$

$$\dot{p} = \left.\frac{\mathrm{d}p}{\mathrm{d}t}\right|_{\text{adiabatic}} \tag{9}$$

where the change of temperature and pressure due to the adiabatic cooling and expansion is included. As in former studies (Spreitzer et al., 2017; Baumgartner and Spichtinger, 2019), constant temperature and pressure is assumed. The tendencies of temperature and pressure are only preserved in the saturation equation. This means that the equations (8) and (9) are removed and temperature and pressure are set to fixed parameters $T = T_{\text{env}}, p = p_{\text{env}}$. Actually, temperature and pressure changes during a nucleation event are small and their neglect in the growth and nucleation rates enables us to control the impact of these different processes.

Similarly as in the approach in Kärcher and Lohmann (2002), the impact of latent heat release is neglected. As discussed in appendix B, for high temperatures $T > 230\,\text{K}$ this leads to a change in the saturation ratio and thus an enhanced ice crystal number concentration in comparison to the results of the full system. This has to be considered for comparing the results of nucleation events with the "benchmark" simulations by Kärcher and Lohmann (2002).

The nucleation term can be described as

$$\text{Nuc}_n = J_{\text{nuc}} V_d n_a, \quad \text{Nuc}_q = m_0 \text{Nuc}_n \tag{10}$$





where, $V_d$ is the mean volume of a supercooled solution droplet, $n_a$ is the number concentration of solution droplets, and $m_0$ is the mean mass of a newly frozen solution droplet, which can be set to $m_0 = 10^{-16}\,\mathrm{kg}$. The nucleation rate for the homogeneous freezing of solution droplets is denoted by $J_{\mathrm{nuc}}$. For comparison with former investigations (Kärcher and Lohmann, 2002; Spichtinger and Gierens, 2009), we set the number concentration of the background aerosol to a quite large value of $n_a\rho = 10^4\,\mathrm{cm}^{-3} = 10^{10}\,\mathrm{m}^{-3}$. We will later discuss the impact of this value in terms of nucleation events.

The diffusional growth of ice crystals is determined by the growth rate

$$\mathrm{Dep}_q = n_i \cdot 4\pi D_v^* C G_v (S_i - 1) f_v \tag{11}$$

with the diffusion constant for water vapour in air $D_v^* = D_v(p,T) f_D$ as corrected by the factor $f_D$ for the kinetic regime, the capacity of ice crystals, $C$, assuming columnar shape, the Howell factor $G_v(p,T)$ describing the impact of latent heat, and the ventilation correction $f_v$, respectively. Note, that the capacity also depends on the mean mass of the ice crystal ensemble, i.e. $C = C(\bar{m}) = C(n_i, q_i)$. The details of the formulation are given in appendix A.

For the determination of the source and sink terms for the supersaturation, we investigate the total derivative of the saturation ratio $S_i = \frac{p\,q_v}{\varepsilon_0 p_{si}(T)}$, where $\varepsilon_0$ denotes the ratio of molar masses of water and dry air. Within this computation, we incorporate the tendencies for temperature and pressure. The temperature tendency is separated into an adiabatic and a diabatic contribution

$$\begin{aligned} \left.\frac{\mathrm{d}T}{\mathrm{d}t}\right|_{\mathrm{adiabatic}} &= -\frac{g}{c_p} w, \\ \left.\frac{\mathrm{d}T}{\mathrm{d}t}\right|_{\mathrm{diabatic}} &= \frac{L}{c_p} \left.\frac{\mathrm{d}q_i}{\mathrm{d}t}\right|_{\mathrm{phase}} = \frac{L}{c_p} \left(\mathrm{Nuc}_q + \mathrm{Dep}_q\right), \end{aligned} \tag{12}$$

and the expression

$$\left.\frac{\mathrm{d}p}{\mathrm{d}t}\right|_{\mathrm{adiabatic}} = -g\rho w \tag{13}$$

describes the pressure tendency, respectively. In these equations, $w$ denotes the vertical velocity of the air parcel and $c_p$ is the specific heat capacity of dry air (assumed as a constant, see, Baumgartner et al., 2020). These terms lead to the source and sink terms for $S_i$, i.e.,

$$\mathrm{Cool} = \left[\frac{L\,g}{c_p R_v T^2} - \frac{g}{R_a T}\right] S_i w \tag{14}$$

and

$$\mathrm{Dep}_s = -\left[\frac{L^2}{c_p R_v T^2} + \frac{1}{q_v}\right] S_i \left(\mathrm{Nuc}_q + \mathrm{Dep}_q\right) \tag{15}$$

To a good approximation, for cold temperatures the first term in the bracket in (15) can be omitted. This is also consistent with our assumption that during the nucleation event temperature and pressure do not change. Thus, we arrive at

$$\mathrm{Dep}_s \approx -\frac{p}{\varepsilon_0 p_{\mathrm{si}}} \left(\mathrm{Nuc}_q + \mathrm{Dep}_q\right) \tag{16}$$





Combining the expressions from above, the reduced system of equations reads

$$\dot{n}_i = \mathrm{Nuc}_n \tag{17}$$

$$\dot{q}_i = \mathrm{Nuc}_q + \mathrm{Dep}_q \tag{18}$$

$$\dot{S}_i = \left[\frac{Lg}{c_p R_v T^2} - \frac{g}{R_a T}\right] S_i w + \frac{p}{\varepsilon_0 p_{\mathrm{si}}}\left(\mathrm{Nuc}_q + \mathrm{Dep}_q\right) \tag{19}$$

**Remark:** As shown in Spreitzer et al. (2017), it is possible to determine and characterize the steady states of the reduced system, which additionally includes sedimentation. This leads to a nonlinear oscillator with a bifurcation diagram, depending on the updraft velocity $w$, and on the temperature $T$.

## 3 Investigations of the nucleation rates

Investigations of ice clouds in the cold temperature regime ($T < 235\,\mathrm{K}$) need to include the nucleation process of homogeneous freezing of aqueous solution droplets. As pointed out in section 1 the formulation by Koop et al. (2000) based on water activity is a meaningful fit to experimental data. However, for theoretical investigations and the use in reduced order models, a simpler but still accurate approximation would be helpful. In the following we present a way how to derive such an approximation based on the original fit through measurements by Koop et al. (2000) in addition to recent observations for pure super-cooled water.

### 3.1 Correction of the nucleation rate

In the study by Koop and Murray (2016) a parametrisation of the nucleation rate of pure supercooled water $J_{\mathrm{pure\,liq}}(T)$ was derived, based on recent measurements. Thus, in the context of homogeneous freezing of solution droplets, the nucleation rate for pure water particles should coincide with the nucleation rate of solution droplets $J_{\mathrm{sol}}(\Delta a_w)$ at water saturation, i.e. the condition

$$J_{\mathrm{sol}}(\Delta a_w) \overset{RH=1}{\equiv} J_{\mathrm{pure\,liq}}(T) \tag{20}$$

should hold. However, evaluating these two formulations of the nucleation rates at water saturation shows nonequal values. A reasonable requirement is that the values of both formulations should match in the temperature range $235\,\mathrm{K} \le T \le 240\,\mathrm{K}$, since this range is relevant for the freezing of pure water cloud droplets with reasonable sizes. This temperature range at water saturation is equivalent to the range of water activity difference $0.27 \le \Delta a_w \le 0.31$. The offset between the curves is shown in figure 1 and can be corrected by shifting the nucleation rate for solution droplets by a constant value. The value of the shift was calculated by minimising the square distance between the curves in the respective temperature range. Thus, the corrected nucleation rate for aqueous solution droplets reads as

$$\log_{10}(J_{\mathrm{sol,new}}(\Delta a_w)) = \log_{10}(J_{\mathrm{sol}}(\Delta a_w)) + \delta \tag{21}$$

with $\delta = -1.522$. The nucleation rates are given in SI units (as used for all quantities throughout this study), i.e. $[J] = \mathrm{m}^{-3}\,\mathrm{s}^{-1}$.
**Remarks:**



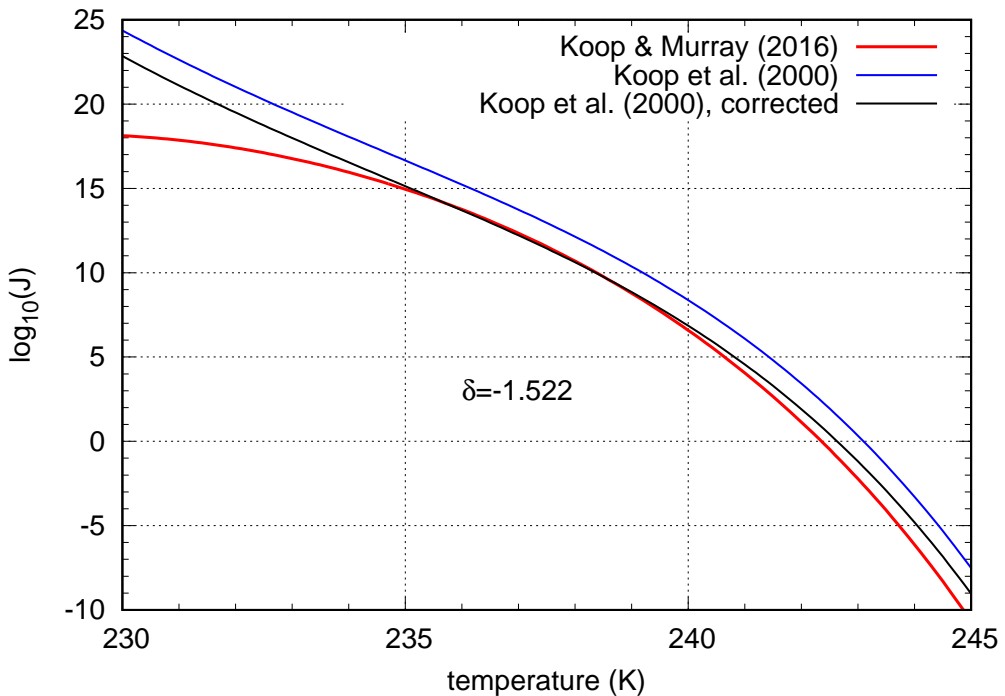

**Figure 1.** Nucleation rates for pure super-cooled water droplets (Koop and Murray, 2016, red) and aqueous solution droplets (Koop et al., 2000); original values in blue, shifted values ($\delta = -1.522$) in black (new reference nucleation rate $J_{\mathrm{sol,new}}$).

– The nucleation rate of pure water droplets can be used for a direct parametrisation of the nucleation rate of aqueous
solution droplets. This will be carried out in section 5.1.

– In the following we will refer to the corrected nucleation rate as "reference" nucleation rate, since, to the best of our
knowledge, it provides the best and most recent fit for the homogenous nucleation rate of solution particles.

### 3.2 Nucleation rate as a function of $T$ and $S_i$

The general strategy of the study is to represent the exponent of the nucleation rate by low order polynomials in a thermody-
namic variable $x$, i.e.

$$J = 10^{p(x)}, \ \ p(x) = \sum_{k=0}^{n} a_k x^k, \ \ \deg p = n. \tag{22}$$

For instance, the formulation of the nucleation rate for aqueous solution droplets by Koop et al. (2000) is based on a polynomial
of degree three, i.e.

$$J_{\mathrm{sol}}(\Delta a_w) = 10^{p_3(\Delta a_w)}, \ \ p_3(x) = \sum_{k=0}^{3} a_k x^k \tag{23}$$





using the thermodynamic quantity $x = \Delta a_w = a_w - a_w^i$.

Note, also the nucleation rate $J_{\text{pure liq}}$ for pure water droplets is based on the same structure, i.e. $\log_{10}(J_{\text{pure liq}})$ is a polynomial of order 6 in the thermodynamic variable $T$ (cf. Koop and Murray, 2016). For analytical investigations of the homogeneous nucleation, it is desirable to represent $\log_{10}(J)$ by a polynomial with low degree. As will be shown in the following, the formulation

$$\log_{10}(J) \approx p(x), \ \deg p \leq 2 \tag{24}$$

with a polynomial yields sufficient agreement with the reference. For analytical investigations (e.g. using asymptotic analysis) it is helpful to represent the nucleation rate using a threshold for the humidity to account for the explosive character of nucleation events as used in the analysis by Baumgartner and Spichtinger (2019). Thus, for the nucleation rate for super-cooled solution droplets we make the following ansatz

$$J = 10^{p(x)} \sim \exp\left(A(T)(S_i - S_c) + B(T)(S_i - S_c)^2\right) \tag{25}$$

where $S_c = S_c(T)$ is the temperature-dependent threshold value for the saturation ratio. In order to describe $J$ as a function of $S_i$ and $T$ we reformulate $\Delta a_w$ as

$$\Delta a_w = (S_i - 1)a_w^i(T) = (S_i - S_c)a_w^i(T) + (S_c - 1)a_w^i(T) \tag{26}$$

using a threshold $S_c(T)$; it corresponds to a fixed value $J_0$ of the nucleation rate, i.e. $J(S_c(T), T) = J_0$. Taking the logarithm,

this equality implies $p(x_0) = j_0 = \log_{10}(J_0)$ with $\Delta a_w = x_0$. As in former studies (see, e.g., Koop et al., 2000; Kärcher and Lohmann, 2002) , we choose $J_0 = 10^{16}\,\text{m}^{-3}\,\text{s}^{-1} = 10^{10}\,\text{cm}^{-3}\,\text{s}^{-1}$. Note, that this choice is quite arbitrary and has no strict physical explanation. Evaluating eq. (26) at $S_i = S_c$, we arrive at

$$p^{-1}(j_0) = x_0 = (S_i - S_c)a_w^i(T) + (S_c - 1)a_w^i(T)$$
$$\stackrel{S_i = S_c}{=} (S_c - 1)a_w^i(T) \tag{27}$$

leading to a description of the threshold

$$S_c = \frac{1}{a_w^i(T)}p^{-1}(j_0) + 1 \quad \text{and}$$
$$\Delta a_w = (S_i - S_c)a_w^i(T) + p^{-1}(j_0) \tag{28}$$

if the polynomial $p(x)$ can be inverted in the relevant range $0.26 \leq \Delta a_w \leq 0.34$. Combining the equations from above, the nucleation rate can be represented as

$$\log_{10} J = p(\Delta a_w) = p\left((S_i - S_c)a_w^i(T) + p^{-1}(j_0)\right) \tag{29}$$

which is a threshold description using the thermodynamic variables $S_i, T$. This representation amounts to a reformulation of

the original approximation, if the inverse function $p^{-1}(x)$ exists in the relevant range (i.e. $p(x)$ is strictly monotonic). In the following we consider linear and quadratic polynomials.





1. Case of a linear polynomial $p_1(x) = a_0 + a_1 x$

   The inverse function of $p_1(x) = y$ is given by $p_1^{-1}(y) = \frac{y-a_0}{a_1}$ implying the threshold

$$S_c(T) = \frac{1}{a_w^i(T)} \frac{j_0 - a_0}{a_1} + 1. \tag{30}$$

Substituting eq. (30) into the expression (29) yields

$$\begin{aligned} \log_{10} J(S_i, T) = j(S_i, T) &= j_0 + a_1 a_w^i(T)(S_i - S_c(T)) \\ &= j_0 + A(T)(S_i - S_c(T)) \end{aligned} \tag{31}$$

   using the approximation $p_3(\Delta a_w) \approx p_1(\Delta a_w) = a_0 + a_1 \Delta a_w$ whereas $A(T) = a_1 a_w^i(T)$; the coeffifients $a_0, a_1$ can be determined in different ways, see section 3.3. Furthermore, approximations to the functions $A(T)$ and $S_c(T)$ can be investigated.

2. Case of quadratic polynomial $p_2(x) = b_0 + b_1 x + b_2 x^2 = a(x - b)^2 + c$

   Since a quadratic function is not strictly monotonic in general, inverting the quadratic polynomial leads to two functions, i.e.

$$p_2^{-1}(y) = b \pm \sqrt{\frac{y - c}{a}} \tag{32}$$

   If one solution can be ruled out (e.g. due to physical constraints) we can formulate

$$\log_{10} J = p_2\left((S_i - S_c(T))a_w^i(T) + p_2^{-1}(j_0)\right) \tag{33}$$

   using the threshold description

$$S_c(T) = \left(b \pm \sqrt{\frac{j_0 - c}{a}}\right) \frac{1}{a_w^i(T)} + 1 \tag{34}$$

   Equivalently, we can derive a formulation

$$\begin{aligned} \log_{10} J = c_0 &+ q_1(T)(S_i - S_c(T)) \\ &+ q_2(T)(S_i - S_c(T))^2 \end{aligned} \tag{35}$$

with appropriate functions $q_1, q_2$, which might be useful for analytic investigations.

   **Remark:** We will use this quadratic ansatz for a direct approximation of the nucleation rate of pure water droplets (see section 5.1).

## 3.3 Linear polynomial fit for the nucleation rate

In this section we investigate approximations of the exponent of the nucleation rate of aqueous solution droplets $J_{\mathrm{sol}}$ and their
impact on nucleation events in an idealised scenario. We concentrate on the reference formulation (Koop et al., 2000). Since




the polynomial $p_3(x)$ in the original formulation

$$J_{\text{sol}}(\Delta a_w) = 10^{p_3(\Delta a_w)}, \quad p_3(x) = \sum_{k=0}^{3} a_k x^k \tag{36}$$

nearly behaves as a linear polynomial in the relevant range $0.26 \leq \Delta a_w \leq 0.34$, it can be easily approximated by a linear relation, i.e. $p_3(x) \approx b_0 + b_1 x$. For this we can use two different approaches: (i) using a least square fit to $p_3$ and (ii) a Taylor expansion at a prescribed value $y_0$. While the first approach is just a fitting procedure in the relevant range $0.26 \leq \Delta a_w \leq 0.34$, the second approach relies on an a priori choice for the evaluation point $y_0 \in [0.26, 0.34]$; it is not evident from the outset which value should be used to provide an accurate approximation. For this, we investigate the sensitivity of $p_3$ to a small perturbation $\varepsilon = y - y_0$, i.e. we consider

$$p_3(y) = p_3(y_0 + \varepsilon) \quad = \quad p_3(y_0) + \left.\frac{\mathrm{d}p_3}{\mathrm{d}x}\right|_{y_0} \varepsilon + \mathcal{O}\left(\varepsilon^2\right) \tag{37}$$

$$\approx \quad b_{t0} + b_{t1} \cdot y = p_{t,y_0}(y) \tag{38}$$

with the coefficients

$$b_{t0} = p_3(y_0) - \left.\frac{\mathrm{d}p_3}{\mathrm{d}x}\right|_{y_0} \cdot y_0 \quad \text{and} \quad b_{t1} = \left.\frac{\mathrm{d}p_3}{\mathrm{d}x}\right|_{y_0}. \tag{39}$$

The Taylor approximation leads us to a range for the slope of the linear approximation; these values motivate later the sensitivity analysis in section 3.5.2. In the relevant range $0.26 \leq y \leq 0.34$ for $y = \Delta a_w$ we obtain slopes in the range $221 \leq b_{t1} \leq 453$. This investigation gives us a hint about possible variations in the slope of $p_{ls}(x)$, which will be used later for the sensitivity analysis in section 3.5.2.

In contrast, using a least square fitting routine for $0.26 \leq \Delta a_w \leq 0.34$ we obtain a linear function

$$p_{ls}(x) = b_{ls,0} + b_{ls,1} \cdot x \tag{40}$$

with $b_{ls,0} = -62.19267$ and $b_{ls,1} = 254.7749$. For the further investigations, we only use the linear fit from eq. (40).

We observe that the linear fit $p_{ls}(x)$ best approximates $p_3$ close to the inflection point $x_{\text{infl}} \approx 0.30756$ (see figure 2, left panel). For each linear approximation $p(x) = b_0 + b_1 \cdot x$ of $p_3(x)$, the exponent of the nucleation rate and the saturation ratio threshold become, as demonstrated in section 3.2,

$$j(S_i, T) = j_0 + \underbrace{b_1 a_w^i(T)}_{:=A(T)}(S_i - S_c(T)),$$

$$S_c(T) = \frac{1}{a_w^i(T)} \frac{j_0 - b_0}{b_1} + 1. \tag{41}$$

Since $a_w^i$ is a rather complicated function of temperature, it is particularly useful in the context of analytical investigations to have simpler approximations of this quantity. This motivates to approximate $a_w^i$ and its inverse $\frac{1}{a_w^i}$ in the relevant temperature range $190 \leq T \leq 230\,\text{K}$ by polynomials $q(T)$ of degree $\deg q \leq 2$. Similarly, we can approximate the nucleation threshold $S_c(T)$ by polynomials $s(T)$ of degree $\deg s \leq 2$. For the approximation we use a least square procedure within the temperature



range $190 \leq T \leq 230\,\mathrm{K}$. The results are presented in figure 2 (middle and right panels). Note that the thresholds, either exact or approximate, are quite similar to the former approximations by Ren and Mackenzie (2005), while there is a larger difference

to the approximation by Kärcher and Lohmann (2002).

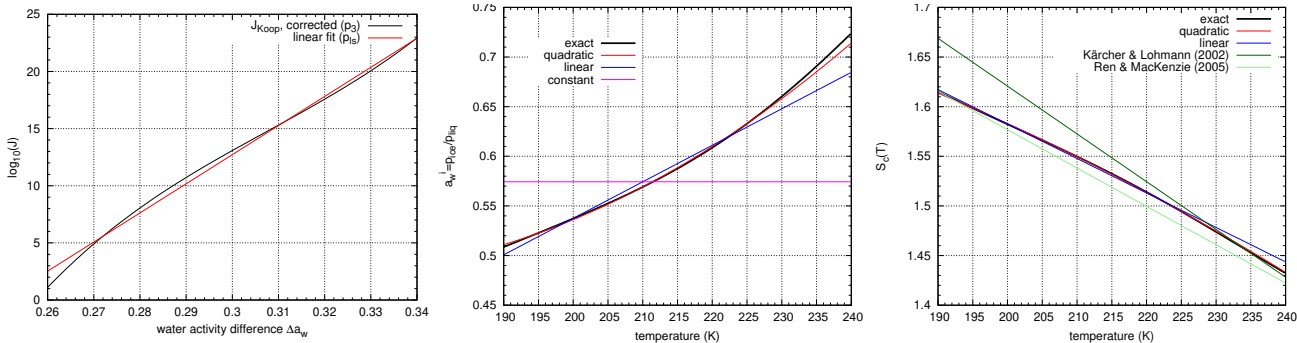

**Figure 2.** Polynomial approximations of the nucleation rate (left), the ice water activity $a_w^i(T) = \frac{p_{\mathrm{ice}}(T)}{p_{\mathrm{liq}}(T)}$ (middle), and the saturation ratio threshold $S_c(T)$ (right panel), respectively. The right panel also includes the approximations by Kärcher and Lohmann (2002); Ren and Mackenzie (2005)

Combining the approximations $q(T)$ and $s(T)$ yields the formulation

$$
\begin{aligned}
j(S_i,T) &= j_0 + b_1 q(T)(S_i - s(T)) \\
&\approx j_0 + A(T)(S_i - S_c(T))
\end{aligned}
\tag{42}
$$

of $\log_{10}(J)$. As can be seen in Figure 2, the nucleation threshold is accurately approximated by a linear relation (deviation is smaller than $0.3\%$). In former studies (e.g. Kärcher and Lohmann, 2002; Ren and Mackenzie, 2005) linear fits were derived for

the nucleation thresholds; however, these fits deviate significantly more from the reference in comparison to ours (see figure 2). The deviation depends on the respective formulation (or approximation) of $a_w^i$.

### 3.4 Thresholds for prescribed nucleation rate values

The threshold description in section 3.3 was based on the (arbitrary) choice of a nucleation rate value $j_0 = 16$, leading to a nucleation rate $J = 10^{16}\,\mathrm{m}^{-3}\mathrm{s}^{-1}$. As already mentioned, the choice of $j_0$ is quite arbitrary, and these high values of $J$ are

very often not reached in the numerical simulations (see section 3.5). For a better diagnostics of the nucleation events and the relative strength of nucleation events, we introduce a similar concept for nucleation thresholds, based on a prescribed nucleation rate value $J \sim 10^{x_0}$. For this purpose we repeat eq. (42) of the nucleation threshold based on the linear approximation of the nucleation rate:

$$
j(S_i,T) = j_0 + A(T)\,(S_i - S_c(T))
\tag{43}
$$





With a fixed but arbitrary value $x_0 > 0$, we can write

$$
\begin{aligned}
x_0 = j(S_0, T) &= j_0 + A(T)\left(S_0 - S_c(T)\right) \\
\Leftrightarrow\ S_{cx0}(T) = S_0 &= \frac{x_0 - j_0}{A(T)} + S_c(T)
\end{aligned}
\tag{44}
$$

whereas the function $A(T) = b_1 a_w^i(T)$ depends only on the linear approximation of $J$ as stated in section 3.2. Note that obviously $S_{cx0}(T) = S_c(T)$ for $x_0 = j_0$. This leads to the formulation of the nucleation rate

$$
j(S_i, T) = x_0 + A(T)\left(S_0 - S_{cx0}(T)\right).
\tag{45}
$$

with a general nucleation value $x_0$ and its associated threshold function $S_{cx0}(T)$. The threshold function is just constantly shifted on the vertical axis, i.e. the type of the threshold function remains the same. This formulation will be used for the theoretical investigations using small perturbations (see section 3.6)

### 3.5 Numerical simulations of nucleation events for different approximations

In the following we will investigate the impact of our approximations of $\log_{10}(J)$ on nucleation events. The setup is as follows:
We use the simple bulk ice physics model as described by the set of ODEs (17), (18), (19) in section 2. A nucleation event is ensured by assuming a constant vertical velocity, which directly translates into an adiabatic cooling of the air parcel and, thus, an initially increasing saturation ratio. Note, we do not change the temperature directly, but instead use the term $\frac{\mathrm{d}T}{\mathrm{d}t} = -\frac{g}{c_p} w$ as a forcing of supersaturation as indicated in eq. (19). This approach was already successfully employed in Spreitzer et al. (2017) and allows to control the nucleation event without the need to disentangle the different contributions of temperature and
supersaturation.

In the reference study by Kärcher and Lohmann (2002) we ignore the impact of latent heat in the simple model, but this effect was included in the study by Spichtinger and Gierens (2009). As described in appendix B, the impact of latent heat is negligible for temperatures $T < 230\,\mathrm{K}$. For higher temperatures, nucleation parameterisations based on Kärcher and Lohmann (2002) might lead to higher ice crystal concentrations in comparison to formulations including latent heat release.

Idealised nucleation events have always the same structure: Due to the supersaturation source $\sim w S_i$ with constant updraft $w$ the variable $S_i$ increases and the nucleation term produces ice crystals, which can grow by water vapour diffusion, constituting a sink for supersaturation. The peak value of $S_i$ is reached at balance between source and sink of supersaturation, after this maximum the variable $S_i$ decreases due to diffusional growth and thus shut off the nucleation term. The peak value depends crucially on the number of nucleated ice crystals that are needed, to balance the source for $S_i$ by the diffusional growth
(depending on the product of number concentration and mean radius of ice crystals). The number concentration of ice crystals produced in the nucleation event clearly depends on the vertical velocity $w$ (source term) and the environmental conditions (diffusion depends on temperature and pressure). For details of the time evolution of nucleation events see appendix A.

### 3.5.1 Standard approximation

We compare the following four different representations of the nucleation rates using numerical simulations:





1. nucleation rate in the water activity formulation by Koop et al. (2000) with the correction as described in section 3.1
(reference nucleation rate)

2. water activity approximated by the linear fit as described in section 3.3 (see eq. (40), linear regression)

3. nucleation rate as a function of $S_i, T$ as described in section 3.2 based on the formulation

$$\log_{10} J = j_0 + A(T)(S_i - S_c(T)) \tag{46}$$

of the exponent of the nucleation rate. We compare the following two sets of approximations for $A(T)$ and $S_c(T)$,
respectively:

(a) a linear approximation for $A(T)$ and a quadratic approximation for $S_c(T)$, and

(b) a constant approximation for $A(T)$ and a linear approximation for $S_c(T)$.

These are specific cases, however arbitrary combinations of approximations for $A(T)$ and $S_c(T)$ might be used.

Figure 3 shows the approximated exponents of the nucleation rate together with the (corrected) reference formulation by
Koop et al. (2000) for the three standard temperatures $T = 196, 216, 236\,\mathrm{K}$ as functions of $\Delta a_w$. These temperatures are chosen
for direct comparison with former studies (Kärcher and Lohmann, 2002; Spichtinger and Gierens, 2009). Evidently, the linear
fit with respect to water activity is very close to the reference, and the same is true for the case of a linear function $A(T)$ and
a quadratic approximation $S_c(T)$. For the simplest approximation (constant function $A(T)$ and linear approximation $S_c(T)$),
larger deviations from the reference nucleation rate can be seen.

For $T = 196\,\mathrm{K}$, there is a strong underestimation in the lower range of $\Delta a_w$, whereas for $T = 236\,\mathrm{K}$ the underestimation is
most pronounced for higher values of $\Delta a_w$. In both cases, we expect deviations in the number concentrations of nucleated ice
crystals during the nucleation event and the maximum saturation ratio attained.

We investigate standard nucleation events in terms of (i) the resulting ice crystal number concentration as in former studies
(e.g. Kärcher and Lohmann, 2002; Spichtinger and Gierens, 2009) and of (ii) the maximum (peak) supersaturation, which was
reached during the nucleation event. The latter is usually not investigated; however, for comparisons with real measurements,
e.g. in cloud chambers, these values are of interest.

Figure 4 shows the results of the numerical simulations, i.e. the number of nucleated ice crystals (left panel) and the max-
imum saturation ratio (right panel) at environmental pressure $200\,\mathrm{hPa}$ (the results are similar for other environmental condi-
tions).

Comparing the number of nucleated ice crystals as well as the maximum saturation ratio it is evident, that the difference
between the reference calculation, based on the corrected nucleation rate by Koop et al. (2000), and the runs using the approx-
imated nucleation rates are rather small.

For most nucleation events, the deviation from the reference simulations is not larger than $\pm 15\%$. To assess these deviations
one should keep in mind that measurements of ice crystal number concentrations are quite difficult and the uncertainties
are usually larger than $15\%$. For instance, for the FSSP instrument, which was used in many flight campaigns (e.g. Voigt





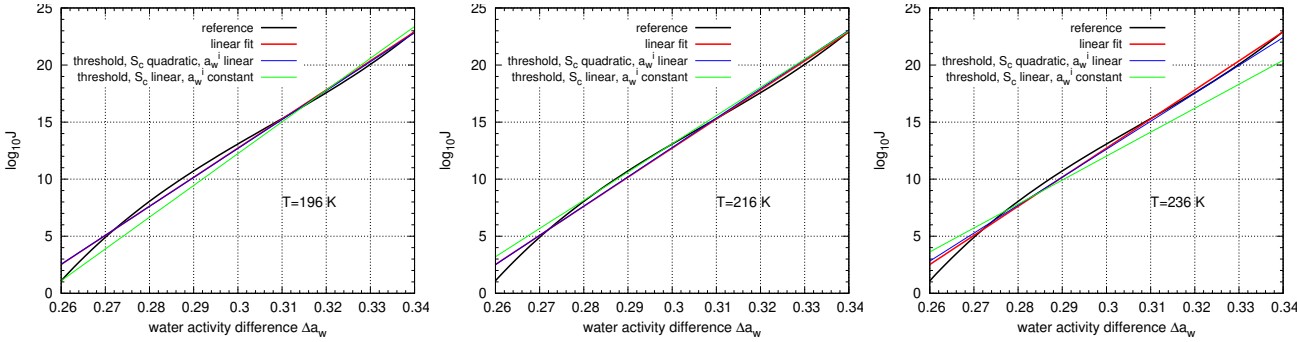

**Figure 3.** Different approximations of nucleation rate for different temperatures (left: $T = 196\,\mathrm{K}$, middle: $T = 216\,\mathrm{K}$, right: $T = 236\,\mathrm{K}$). Black: Reference nucleation rate; red: linear fit to reference nucleation rate; blue: threshold description due to eq 46, using a linear approximation for $a_w^i$ and a quadratic threshold function $S_c$; green: threshold description due to eq. (46), using a constant for $a_w^i$ and a linear threshold function $S_c$.

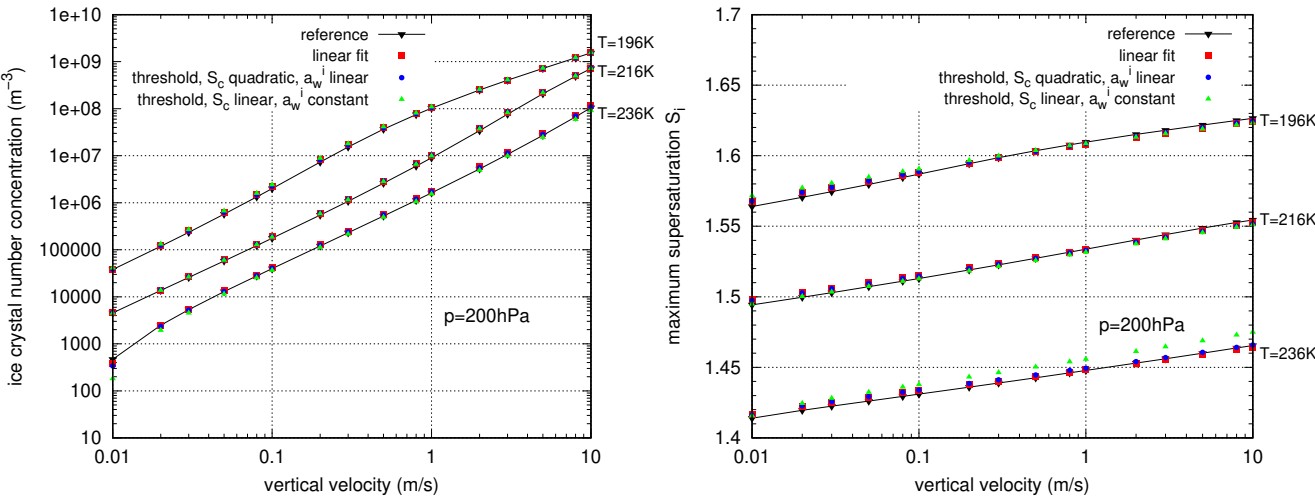

**Figure 4.** Comparison of different approximations of the nucleation rate by Koop et al. (2000) for standard nucleation events as driven by constant vertical velocity $w$. Left: ice crystal number concentration; right: maximum supersaturation





et al., 2017), the uncertainty is estimated by about $\sim 10\%$ (de Reus et al., 2009). Thus, the deviations in our simulations and the uncertainties of realistic measurements are roughly of the same order. This fact renders it presumably impossible to decide on the correctness of any of the different formulations and approximations of the nucleation rate based on the available
observations.

Finally, we conclude that a linear approximation of the reference nucleation rate by Koop et al. (2000) is accurate enough to represent nucleation events in a physically meaningful way. Thus, we can use this description as well as the derived formulations of the nucleation rate as a function of temperature $T$ and saturation ratio $S_i$ in order to investigate which parameters of the nucleation rates significantly affect the outcome of nucleation events. This will be carried out in the next section.

### 350 3.5.2 Impact of the parameters of the linear approximation

In the following, we investigate the influence of the parameters of the linear approximation of $\log_{10}(J)$ in a more qualitative way. In the preceding section we already showed the good approximation quality of the linear approximation. Thus, we use the linear representation $\log_{10} J = b_0 + b_1 \cdot \Delta a_w$ in order to test the sensitivity of nucleation events on the two parameters $b_0, b_1$. Parameter $b_0$ controls the absolute value of the nucleation rate while parameter $b_1$ accounts for its steepness.

In a first step, we investigate the impact of the steepness of the nucleation rate, i.e. the influence of the coefficient $b_1$. One should keep in mind that during the nucleation event the value of $\Delta a_w = (S_i - 1)a_w^i(T)$ is increasing as $S_i$ increases, thus the exponent of the nucleation rate basically grows linearly. Consequently, an increase in the saturation ratio immediately translates into an increase in $\Delta a_w$, hence the abscissa in figure 5 may be thought of as representing saturation ratio. If high values of the nucleation rate are already reached at lower supersaturation values, the nucleation is triggered earlier in comparison to the
reference scenario.

However, an earlier onset of ice nucleation implies that the newly nucleated ice crystals already start to grow by diffusion. As a consequence, the growing ice crystals tend to decrease the saturation ratio and, if they are sufficiently numerous, prematurely stop the ice nucleation event. In this case, less ice crystals will nucleate and a smaller maximum saturation ratio will be reached compared to the reference. The opposite mechanism is expected for smaller values of the nucleation ratio in comparison to the
reference, i.e. higher ice crystals concentrations will occur together with larger maximal saturation ratio.

In order to illustrate this mechanism more quantitatively, we changed the slope of the linear function. The slope is either reduced to a value of $b_1 = 100$ or enhanced to a value $b_1 = 500$, which is motivated by the values of the Taylor approximation, as derived in section 3.3. Note that the "reference" value for the linear fit is $b_1 \approx 255$. In both cases, the parameter $b_0$ of the linear function is adapted such that the inflection point of the polynomial $p_3(\Delta a_w)$ at $\Delta a_w \sim 0.311$ is met for better comparison
with the reference simulations. The resulting nucleation rates are displayed in figure 5, while the number of nucleated ice crystals and the maximum ice saturation ratio during the nucleation event are summarized in figure 6: The left panel shows the concentrations of nucleated ice crystals and the right panel the maximum saturation ratio during the nucleation events.

In case of the enhanced or reduced slope as indicated in figure 5 we exactly see the theoretically proposed behaviour in the ice crystal number concentration; the values are reduced for reduced slopes, and enhanced for enhanced slopes, respectively.



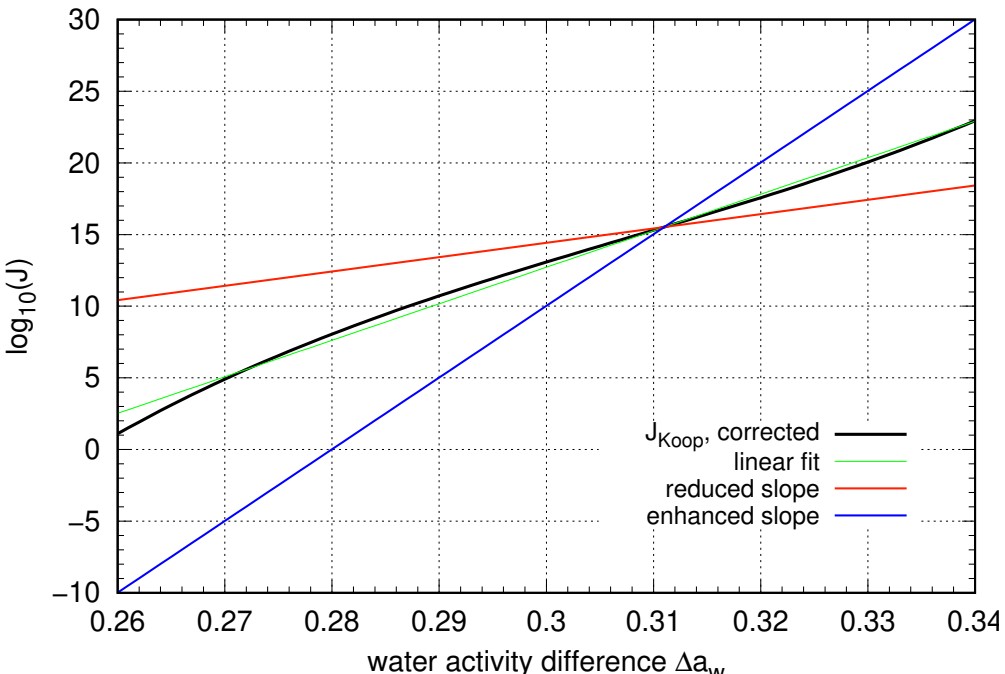

**Figure 5.** Artificial change in the slope of the linear function in the exponent of the nucleation rate. The fit to the reference curve is indicated by the green line (slope $b_1 \sim 250$); a reduced slope ($b_1 \sim 100$) is displayed in red, an enhanced slope ($b_1 \sim 500$) is displayed as blue curve.

The reductions are by up to a factor of $0.4$, the enhancements are by up to a factor of $2.4$, and the largest changes can be seen at the highest temperature $T = 236 \, \mathrm{K}$.

In the right panel of Figure 6, a dependency on temperature and vertical velocity is seen. For very low vertical velocities, the maximum supersaturation behaves as expected, i.e. reduced values for the reduced slope and enhanced values for the enhanced slope, respectively. For very high vertical velocities, this behaviour is reversed, i.e. we see reduced values of $S_{i,\mathrm{max}}$ for enhanced
slopes and enhanced values of $S_{i,\mathrm{max}}$ for reduced slopes, respectively. The transition slightly depends on the temperature. This can be explained as follows: For low vertical velocities, $\Delta a_w$ (and thus the supersaturation) is always below the inflection point $\Delta a_w \sim 0.311$; thus, the nucleation rate is always smaller for the enhanced slope in comparison to the reference while it is always larger in comparison to the reference for the reduced slope. Therefore, in case of an enhanced slope the nucleation starts later compared to the reference. This leads to the behaviour as described above. However, beyond the inflection point
the behaviour is reversed and thus the resulting maximum supersaturation is now enhanced for reduced slope and it is reduced for enhanced slope. The inflection point is reached at different vertical velocities for different temperatures, i.e. for lower





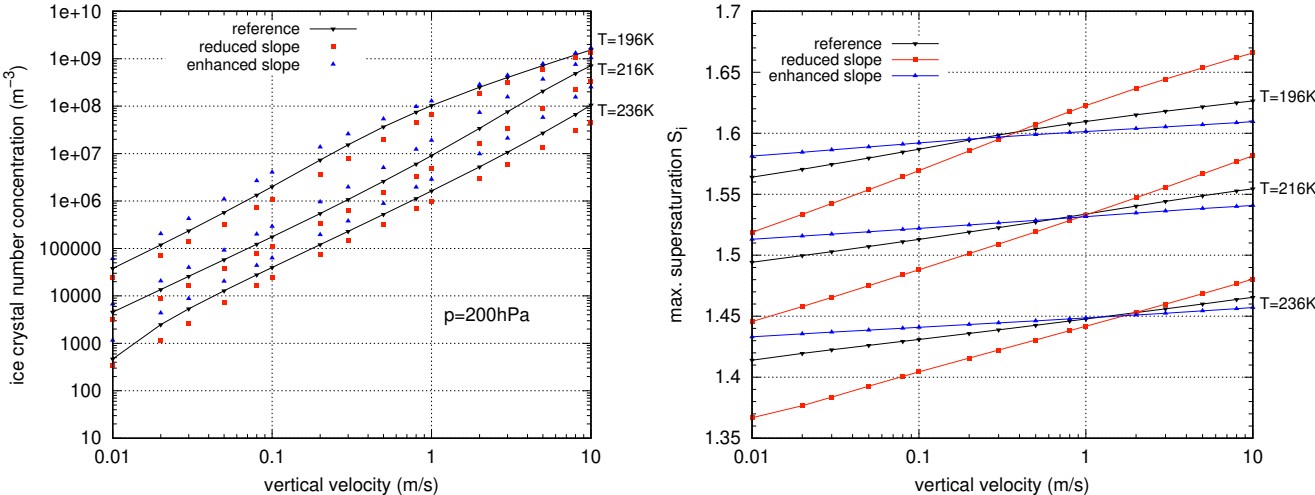

**Figure 6.** Impact of the slope on the idealized nucleation events. Left: ice crystal number concentrations, right: maximum supersaturation values. The colours are chose as in figure 5, i.e. red squares indicate reduced slope, and blue triangles indicate enhanced slope, respectively.

temperatures at lower values of $w$ and for higher temperatures at higher values of $w$. Note, only the maximum supersaturation is affected upon $\Delta a_w$ crossing the inflection point while no influence on the number concentration of ice crystals is seen.

After having varied the slope of the nucleation rate, we now turn to its absolute values and modify coefficient $b_0$, which trans-

lates into a change of values of $J$ by $10^{b_0}$. In order to investigate the sensitivity, we add a constant value $\Delta b \in \{-6, -3, 3, 6\}$ to the coefficient $b_0$, leading to an increase or decrease in the absolute value of the nucleation rate by a factor of $10^{\Delta b}$. In figure 7 the results in terms of ice crystal number concentration and maximum supersaturation are displayed.

Maybe surprisingly, the absolute values of the number concentrations of ice crystals in comparison to the reference formulation are not crucially affected (see figure 7, left panel), although some deviations occur (up to a factor of two). The strongest

deviations can be seen for warm temperatures ($T = 236$ K) at very low vertical velocities. Overall, the relative deviations from the reference events in variables $n_i$ and peak values of $S_i$ are within the interval $[0.4, 2]$, but for vertical velocities in the range $w \geq 0.05\,\mathrm{m\,s^{-1}}$ the relative deviation is within the interval $[0.8, 1.4]$.

Comparing the influence of a scaling of the absolute values of the nucleation rate and the steepness of the rate, we conclude that the correct steepness of the nucleation rate is much more important. Even changes by orders of magnitude in the values

of the nucleation rate has a minor impact on the number of nucleated ice crystals. A similar conclusion was also drawn in the theoretical study by Baumgartner and Spichtinger (2019). In that study, the authors investigated a slightly simplified system of equations by means of asymptotic analysis. The simplified system describes the temporal evolution of the number concentration of ice crystals and the saturation ratio and an approximate asymptotic solution was constructed. To leading order, the approximate solution for the number concentration of ice crystals was completely independent of the precise values of the

nucleation rate, but the steepness contributed directly. The only necessary condition on the values of the nucleation rate was that it attains large values, i.e. significantly larger than the other coefficients within the equations.





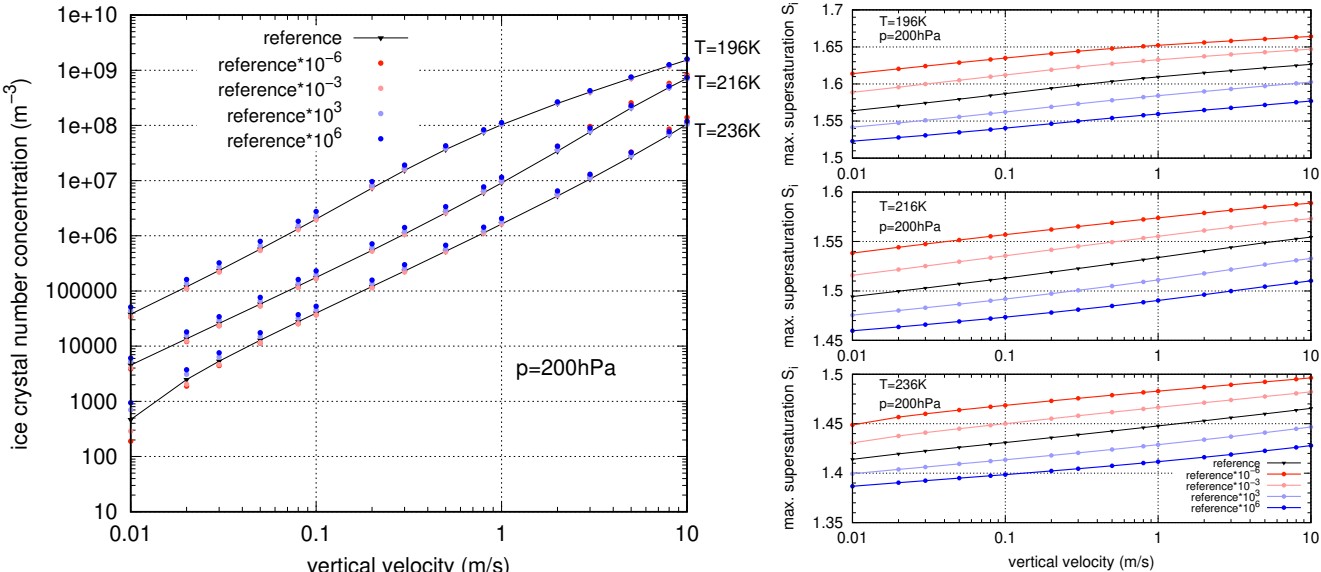

**Figure 7.** Comparison of ice crystal number concentrations (left panel) and maximum supersaturation (right panel) for absolute changes in the nucleation rate by a factor $10^{\Delta b}$, $\Delta b = -6, -3, 3, 6$.

For the maximum supersaturation values, the impact of the absolute value is much more pronounced. As expected, upon reduction of the nucleation rate by a factor of $10^{\Delta b}$ with $\Delta b \in \{-6, -3\}$ the supersaturation reaches much higher values of $S_i$, until the values of the rescaled nucleation rate become large enough to initiate the nucleation of ice crystals. For the

enhancement of the absolute values of the nucleation rate, the results are reversed: The maximum supersaturation is reduced, since the enhanced nucleation rate attains values that allow the production of ice crystals for smaller saturation ratios. This behaviour is represented in the right panel of figure 7.

**Remark:** This idealized enhancement of the nucleation rate can also be seen in the connection with the aerosol number concentration $n_a$. A change of $n_a$ by some orders of magnitudes while no changes in $J$ are applied has the same effect as

changing the absolute value of the nucleation rate (or the parameter $b_0$ in the argument of the exponential function). Thus, a strong reduction or enhancement of the available solution droplets will only slightly change the amount of ice crystals in a nucleation event. Therefore, we can conclude that for a meaningful approximation of the nucleation rate the exact number concentration of available aerosols is also not crucial for the strength of the homogeneous nucleation event, but perhaps for the starting time of the event. Including size effects of the solution droplets might additionally change the picture quantitatively

(see, e.g., Baumgartner et al., 2020).

## 3.6 Impact of perturbations in $S_i$ and $T$ on the nucleation rate

In this section the impact of changes in $S_i$ and/or $T$ on the nucleation rate is theoretically investigated. In the real atmosphere, variations of the temperature due to dynamical processes will introduce such changes, e.g. such as from a passing or even





breaking gravity wave. In numerical simulations, these variations (also often called fluctuations) are often artificially introduced
(e.g. Jensen and Pfister, 2004). In any case, the impact of such changes is investigated using perturbation analysis.

As derived earlier, using the linear approximation of the nucleation rate ($b_1 \approx 254.77$) we can rewrite the nucleation rate as

$$J(S_i, T) = J_{\text{unit}} \cdot 10^{j(S_i, T)} \tag{47}$$

with the function

$$j(S_i, T) = j_0 + A(T)\left(S_i - S_c(T)\right) \text{ and } A(T) = b_1 a_w^i(T). \tag{48}$$

We can estimate the usual values of the function $A(T)$ in the temperature range $190\,\text{K} \leq T \leq 230\,\text{K}$ using $0.51 \leq a_w^i(T) \leq 0.66$
such that $129 \leq A(T) \leq 169$. For a very simple but still sufficient constant approximation of $a_w^i(T)$ we can set $a_{w0}^i = 0.574312$
(see fig. 2, pink line) such that $A(T) \approx A_0 = b_1 a_{w0}^i = 146.32$. Finally we can state $A(T) = \mathcal{O}\left(\varepsilon^{-2}\right)$ with the usual approach
$\varepsilon \sim 0.1$, such that we set $A(T) = A^* \varepsilon^{-2}$ with $A^* = \mathcal{O}(1)$ as $\varepsilon \to 0$. For the non-dimensionalization of the threshold function
in the linear approximation $S_c(T) \approx s_0 + s_1 T$ we have to estimate the order of the coefficients for the relevant temperature
range. Using $190\,\text{K} \leq T_{\text{ref}} \leq 230\,\text{K}$ and the definition $T = T_{\text{ref}}\vartheta$ with the nondimensional temperature $\vartheta$, we have

$$S_c(T) = s_c(\vartheta) = s_0 + s_1 T = s_0 + s_1 T_{\text{ref}}\vartheta = \sigma_0 - \sigma_1\vartheta \tag{49}$$

with $\sigma_1 = -s_1 T_{\text{ref}}$. Obviously, $s_0 = \sigma_0 = 2.27697 = \mathcal{O}(1)$ and $0.66 \leq \sigma_1 \leq 0.8$ such that $\sigma_1 = \mathcal{O}(1)$. Using the simplest approximation $A(T) = A_0$ and $S_c(T) = s_0 + s_1 T$ for the general formulation of the threshold function $S_{cx0}$ (cf. eq. (44)) we can
simplify the expression as

$$S_{cx0}(T) = \frac{x_0 - j_0}{A_0} + s_0 + s_1 T = \underbrace{\left(\frac{x_0 - j_0}{A_0} + s_0\right)}_{=:s_{x0}} + s_1 T \tag{50}$$


$$= s_{x0} + s_1 T.$$

Using non-dimensionalization we end up with the following representation

$$S_{cx0}(T) = s_{cx0}(\vartheta) = s_{x0} + s_1 T = \sigma_{x0} - \sigma_1\vartheta, \tag{51}$$

with $\sigma_{x0} = s_{x0}, \sigma_1 = -s_1 T_{\text{ref}}$. Finally, we can estimate with $A_0 = A^* \varepsilon^{-2}$, such that we obtain

$$\sigma_{x0} = (A^*)^{-1}\varepsilon^2(x_0 - j_0) + \sigma_0 = \delta + \sigma_0. \tag{52}$$

Since $j_0 = \mathcal{O}\left(\varepsilon^{-1}\right)$ and $x_0 = \mathcal{O}\left(\varepsilon^\beta\right)$ with $\beta \geq -1$ we find $\sigma_{x0} = \delta + \sigma_0 = \mathcal{O}(\varepsilon) + \mathcal{O}(1) = \mathcal{O}(1)$. After non-dimensionalizing
the argument in the nucleation rate, we can now investigate the response of the nucleation rate due to perturbation (i) in
saturation ratio (i.e. in the same way as the numerical simulations are set up), (ii) in temperature, and (iii) in adiabatic changes
of temperature driving changes in the saturation ratio simultaneously. In reality, almost exclusively case (iii) is relevant.

First, we estimate the increase of $J$ due to variations of $S_i$ at a constant temperature $T = T_{\text{ref}}$. For this purpose we start
at a given value of the saturation ratio $S_i$ which corresponds to a certain threshold $x_0$ via the relation (44). We choose this





value as a reference value $S_{\text{ref}} = S_{cx0}(T_{\text{ref}}) = s_{cx0}(1) = \sigma_0 - \sigma_1$; this corresponds to a reference value of the nucleation rate $J = J_{\text{ref}} = J_{\text{unit}} \cdot 10^{x_0}$ (with $J_{\text{unit}} = 1\,\text{m}^{-3}\,\text{s}^{-1}$). Assuming the expansion

$$S_i = S_0 + \varepsilon S_1 + \varepsilon^2 S_2 + \varepsilon^3 S_3 + \mathcal{O}\left(\varepsilon^4\right) \tag{53}$$

for the saturation ratio where $S_0 = S_{\text{ref}}\sigma_{x0} - \sigma_1$ we investigate the impact of such a perturbation on the exponent $j$. Keeping the temperature fixed as in the numerical simulations we arrive at

$$\begin{aligned}
j(S_i, T) = j(s, t) &= x_0 + A_0\left(S_i - S_{\text{ref}}\right) \\
&= x_0 + A^*\varepsilon^{-2}\left(S_{\text{ref}} + \varepsilon S_1 + \varepsilon^2 S_2 + \varepsilon^3 S_3 + \mathcal{O}\left(\varepsilon^4\right) - S_{\text{ref}}\right) \\
&= x_0 + \varepsilon^{-1}A^*S_1 + A^*S_2 + \varepsilon A^*S_3 + \mathcal{O}\left(\varepsilon^2\right).
\end{aligned} \tag{54}$$

We are interested in the relative change of the nucleation rates $\frac{J(S_i, T)}{J_{\text{ref}}}$, which translates into $j(S_i, 1) - j(S_{\text{ref}}, 1)$. By definition, we have $x_0 = j(S_{\text{ref}}, 1)$, thus we obtain

$$j(S_i, 1) - j(S_{\text{ref}}, 1) = \varepsilon^{-1}A^*S_1 + A^*S_2 + \varepsilon A^*S_3 + \mathcal{O}\left(\varepsilon^2\right). \tag{55}$$

Inspecting eq. (55) it is evident, that a nonzero perturbation term $S_\alpha$ in eq. (53) is connected with the factor $\varepsilon^{\alpha-2}$, hence a change of order $\mathcal{O}\left(\varepsilon^\alpha\right)$ translates into a change of order $\mathcal{O}\left(\varepsilon^{\alpha-2}\right)$ in the exponent of $J$.

Second, we consider perturbations of temperature without changing the saturation ratio, although this might not happen in the atmosphere. Using the approach above with a constant reference value of saturation, i.e. $S_{\text{ref}} = s_{cx0}(1) = \sigma_0 - \sigma_1$ and temperature perturbations $\vartheta = 1 + \varepsilon\vartheta_1 + \varepsilon^2\vartheta_2 + \varepsilon^3\vartheta_3 + \mathcal{O}\left(\varepsilon^4\right)$ we find the following expression:

$$\begin{aligned}
j(s_{\text{ref}}, \vartheta) &= x_0 + A_0\left(S_{\text{ref}}\right. \\
&\quad \left. - \left(\sigma_0 - \sigma_1\left(1 + \varepsilon\vartheta_1 + \varepsilon^2\vartheta_2 + \varepsilon^3\vartheta_3 + \mathcal{O}\left(\varepsilon^4\right)\right)\right)\right) \\
&= x_0 + A^*\varepsilon^{-2}\left(\varepsilon\vartheta_1 + \varepsilon^2\sigma_1\vartheta_2 + \varepsilon^3\sigma_1\vartheta_3 + \mathcal{O}\left(\varepsilon^4\right)\right) \\
&= x_0 + \varepsilon^{-1}A^*\sigma_1\vartheta_1 + A^*\sigma_1\vartheta_2 + \varepsilon A^*\sigma_1\vartheta_3 + \mathcal{O}\left(\varepsilon^2\right)
\end{aligned} \tag{56}$$

The relative change of the nucleation rate is then given by

$$\begin{aligned}
j(S_{\text{ref}}, \vartheta) - j(S_{\text{ref}}, 1) &= x_0 + \varepsilon^{-1}A^*\sigma_1\vartheta_1 \\
&\quad + A^*\sigma_1\vartheta_2 + A^*\sigma_1\vartheta_3\varepsilon + \mathcal{O}\left(\varepsilon^2\right)
\end{aligned} \tag{57}$$

Thus, a temperature perturbation $\vartheta_\alpha$ of order $\mathcal{O}\left(\varepsilon^\alpha\right)$ leads to a relative change in $j$ of order $\mathcal{O}\left(\varepsilon^{\alpha-2}\right)$. Note the sign of the perturbations, which turns into the opposite sign in the change of $j$. Because of the strictly monotonic descrease of the threshold function $S_{cx0}(T)$, a negative temperature change leads to a higher threshold and in turn to a lower nucleation rate at a given saturation ratio.

Instead of perturbing the saturation ratio and the temperature individually, these quantities are connected in the real world. To a good approximation, their joint variation is through an adiabatic change. Therefore, we finally investigate the impact of





adiabatic temperature changes on the saturation ratio and in turn on the nucleation rate. For this purpose we have to consider
the dependence of $S_i$ on adiabatic temperature changes; we start with the source term of the saturation ratio

$$dS_i = \left(\frac{1}{\kappa} - \frac{L}{R_v T}\right) S_i \frac{dT}{T} \tag{58}$$

First we estimate the term $\gamma(T) = \frac{1}{\kappa} - \frac{L}{R_v T}$ for $190\,\mathrm{K} \leq T \leq 230\,\mathrm{K}$, leading to $-28.8 \leq \gamma(T) \leq -23.2$ such that we find
$\gamma(T) = \mathcal{O}\left(\varepsilon^{-1}\right) = \gamma^* \varepsilon^{-1}$ and $\gamma^* \sim -2.5 < 0$. We approximate the total differential with finite differences $\Delta S_i, \Delta T$, i.e.

$$\frac{\Delta S_i}{S_i} = \gamma^* \varepsilon^{-1} \frac{\Delta T}{T} \tag{59}$$

We set as an approximation $S_i = S_{\mathrm{ref}}$ and $T = T_{\mathrm{ref}}$ such that we can set

$$\frac{\Delta S_i}{S_{\mathrm{ref}}} = \varepsilon^l S_l + \mathcal{O}\left(\varepsilon^{l+1}\right) = \mathcal{O}\left(\varepsilon^l\right) \tag{60}$$

with $S_l = \mathcal{O}(1)$. We assume $l \geq 1$ since we do not consider changes of the saturation ratio of order $\mathcal{O}(1)$. The analoguous
expansion for the temperature reads

$$\frac{\Delta T}{T_{\mathrm{ref}}} = \varepsilon^k \vartheta_k + \mathcal{O}\left(\varepsilon^{k+1}\right) = \mathcal{O}\left(\varepsilon^k\right) \quad \text{with } \vartheta_k = \mathcal{O}(1) \ \forall k \geq 1. \tag{61}$$

Combining these expansions, we arrive at

$$\begin{aligned}
\frac{\Delta S_i}{S_{\mathrm{ref}}} &= \gamma^* \varepsilon^{-1} \frac{\Delta T}{T_{\mathrm{ref}}} = \gamma^* \varepsilon^{-1} \left(\varepsilon^k \vartheta_k + \mathcal{O}\left(\varepsilon^{k+1}\right)\right) \\
&= \gamma^* t_k \varepsilon^{k-1} + \mathcal{O}\left(\varepsilon^k\right)
\end{aligned} \tag{62}$$

or equivalently

$$\varepsilon^l S_l = \Delta S_i = S_{\mathrm{ref}} \gamma^* \vartheta_k \varepsilon^{k-1} + \mathcal{O}\left(\varepsilon^k\right). \tag{63}$$

The only non-trivial balance is for $l = k - 1$, i.e.

$$S_{k-1} = S_{\mathrm{ref}} \gamma^* \vartheta_k \quad \Leftrightarrow \quad S_k = S_{\mathrm{ref}} \gamma^* \vartheta_{k+1} \tag{64}$$

Note that $l = k - 1 \geq 1$, i.e. we have to consider $k \geq 2$ for the perturbation of temperature. This is a meaningful restriction
since we are interested in small changes of temperature in the cold temperature regime. Thus, we would not expect adiabatic
temperature changes of order $\mathcal{O}(\varepsilon)$, corresponding to changes of order $\sim 10\,\mathrm{K}$. Thus, we assume an asymptotic expansion

$$\vartheta = 1 + \varepsilon^2 \vartheta_2 + \varepsilon^3 \vartheta_3 + \mathcal{O}\left(\varepsilon^4\right) \tag{65}$$

for the temperature perturbation. We are generally interested in adiabatic expansions due to vertical upward motion, which
in turn leads to descreasing temperatures, hence we conclude $\vartheta_k < 0$ for $k \geq 2$. Since $\gamma^* < 0$, equation (64) leads to positive
changes in the saturation ratio $s_k > 0$ for $\vartheta_k < 0$. Generally, warming due to adiabatic compression can be studied in the same
way by setting $\vartheta_k > 0$.





Now we consider the nucleation rate in the formulation for arbitrary thresholds $x_0$ using $S_{\text{ref}} = \sigma_{x0} - \sigma_1 = s_{cx0}(1)$:

$$
\begin{aligned}
j(S_i, T) = j(S_i, t) &= x_0 + A^* \varepsilon^{-2} \left( S_i - s_{cx0}(\vartheta) \right) \\
&= x_0 + A^* \varepsilon^{-2} \left( S_{\text{ref}} + \varepsilon S_1 + \varepsilon^2 S_2 + \varepsilon^3 S_3 \right. \\
&\quad \left. - \left( \sigma_{x0} - \sigma_1 \left( 1 + \varepsilon^2 \vartheta_2 + \varepsilon^3 \vartheta_3 \right) \right) + \mathcal{O}\left( \varepsilon^4 \right) \right) \\
&= x_0 + A^* \varepsilon^{-2} \left( \varepsilon S_1 + \varepsilon^2 S_2 + \varepsilon^3 S_3 + \varepsilon^2 \sigma_1 \vartheta_2 \right. \\
&\quad \left. + \varepsilon^3 \sigma_1 \vartheta_3 + \mathcal{O}\left( \varepsilon^4 \right) \right) \\
&= x_0 + A^* \varepsilon^{-2} \left( \varepsilon S_{\text{ref}} \gamma^* \vartheta_2 + \varepsilon^2 S_{\text{ref}} \gamma^* \vartheta_3 + \varepsilon^3 S_{\text{ref}} \gamma^* \vartheta_4 \right. \\
&\quad \left. + \varepsilon^2 \sigma_1 \vartheta_2 + \varepsilon^3 \sigma_1 \vartheta_3 + \mathcal{O}\left( \varepsilon^4 \right) \right) \\
&= x_0 + A^* \varepsilon^{-2} \left( \varepsilon S_{\text{ref}} \gamma^* \vartheta_2 + \varepsilon^2 \left( S_{\text{ref}} \gamma^* \vartheta_3 + \sigma_1 \vartheta_2 \right) \right. \\
&\quad \left. + \varepsilon^3 \left( S_{\text{ref}} \gamma^* \vartheta_4 + \sigma_1 \vartheta_3 \right) + \mathcal{O}\left( \varepsilon^4 \right) \right) \\
&= x_0 + A^* S_{\text{ref}} \gamma^* \vartheta_2 \varepsilon^{-1} + A^* \left( S_{\text{ref}} \gamma^* \vartheta_3 + \sigma_1 \vartheta_2 \right) \\
&\quad + A^* \left( s_{\text{ref}} \gamma^* \vartheta_4 + \sigma_1 \vartheta_3 \right) \varepsilon + \mathcal{O}\left( \varepsilon^2 \right).
\end{aligned}
\tag{66}
$$

Thus, for $k \geq 2$ we find terms of the form $A^* \left( s_{\text{ref}} \gamma^* \vartheta_{k+1} + \sigma_1 \vartheta_k \right) \varepsilon^{k-2}$ of order $\mathcal{O}\left( \varepsilon^{k-2} \right)$. Comparing the nucleation rates we find for the relative change

$$
\begin{aligned}
j(S_i, T) - j(S_{\text{ref}}, T_{\text{ref}}) &= j(S_i, t) - j(S_{\text{ref}}, 1) \\
&= A^* S_{\text{ref}} \gamma^* \vartheta_2 \varepsilon^{-1} + A^* \left( S_{\text{ref}} \gamma^* \vartheta_3 + \sigma_1 \vartheta_2 \right) \\
&\quad + A^* \left( S_{\text{ref}} \gamma^* \vartheta_4 + \sigma_1 t_3 \right) \varepsilon + \mathcal{O}\left( \varepsilon^2 \right).
\end{aligned}
\tag{67}
$$

For the relative impact of these terms we use the estimations $\gamma^* < -2.3$ and $\sigma_1 \leq 0.8$. We have to distinguish two scenarios for perturbations $\vartheta_k < 0$:

1. $\vartheta_k < 0$ for all $k \geq 2$. In this case we can assume

   $$
   S_{\text{ref}} \gamma^* \vartheta_{k+1} + \sigma_1 \vartheta_k > 0.
   \tag{68}
   $$

   Therefore, an adiabatic temperature perturbation $\vartheta_k$ of order $\mathcal{O}\left( \varepsilon^k \right)$ $(k \geq 2)$ lead to relative changes in $j$ of order $\mathcal{O}\left( \varepsilon^{k-3} \right)$. Note, that the changes in saturation ratio are always dominant and larger than the changes in the threshold, which changes $j$ by order $\mathcal{O}\left( \varepsilon^{k-2} \right)$ in the opposite direction.

2. $\vartheta_k < 0$ and $\vartheta_{k+1} = 0$ for a distinct $k \geq 2$. In this case, the previously discussed temperature effect can be seen, i.e. the nucleation threshold is changed, leading to a reduction of the nucleation rate exponent. This effect is merely academic, since we have to switch off higher perturbations in temperature, which is quite unlikely.

One should keep in mind that we investigated the relative increase in the exponent of the nucleation rate. A relative change of order $\mathcal{O}\left( \varepsilon^k \right)$ in the exponent translates into a relative change of order $\mathcal{O}\left( \exp\left( \varepsilon^k \right) \right)$ in the nucleation rate $J$, thus ranging over several orders of magnitudes.





## 4 Impact of saturation vapour pressure formulation

### 4.1 New representation of saturation water vapour

In the formulation by Murphy and Koop (2005) the extrapolation of the saturation vapour pressure into the no man's land of water's phase diagram is based on the assumption that the state of amorphous ice is thermodynamically equivalent to super-cooled liquid water. Therefore, the specific heat of liquid water can be extended in the super-cooled regime using measurements of amorphous ice. This leads to the established formulation in Murphy and Koop (2005).

Recently, a new representation of the saturation vapour pressure over super-cooled liquid water was proposed by Nachbar et al. (2019). In this study, the authors consider different states of water in the low temperature range. They conclude that 525 amorphous ice is thermodynamically different from super-cooled water, thus they provide a different extrapolation for the saturation vapour pressure (Nachbar et al., 2019). In figure 8 the two formulations are displayed (left: absolute values, right: ratio of functions).

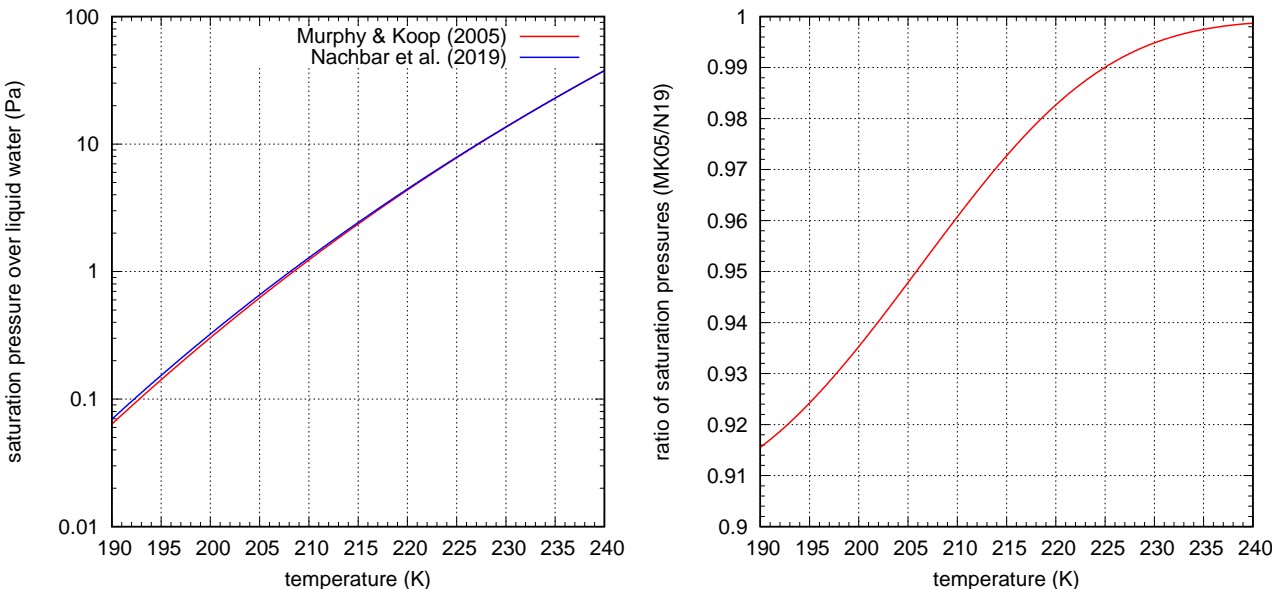

**Figure 8.** Saturation vapour pressure over super-cooled liquid water. Left: two formulations by Murphy and Koop (2005) (red line) and Nachbar et al. (2019) (blue line). Right: ratio of the two formulations $\frac{p_{\text{liq,MK2005}}}{p_{\text{liq,N2019}}}$

Although the deviation between the two curves is very small, even in the low temperature range less than $10\%$, its impact on saturation ratios as well as on the nucleation thresholds is quite large, as can be seen in figure 9.

The curves of water saturation as well as the nucleation thresholds are systematically shifted to higher values. In addition, the new curves have a more linear shape than the curves resulting from Murphy and Koop (2005). The ratio of the saturation



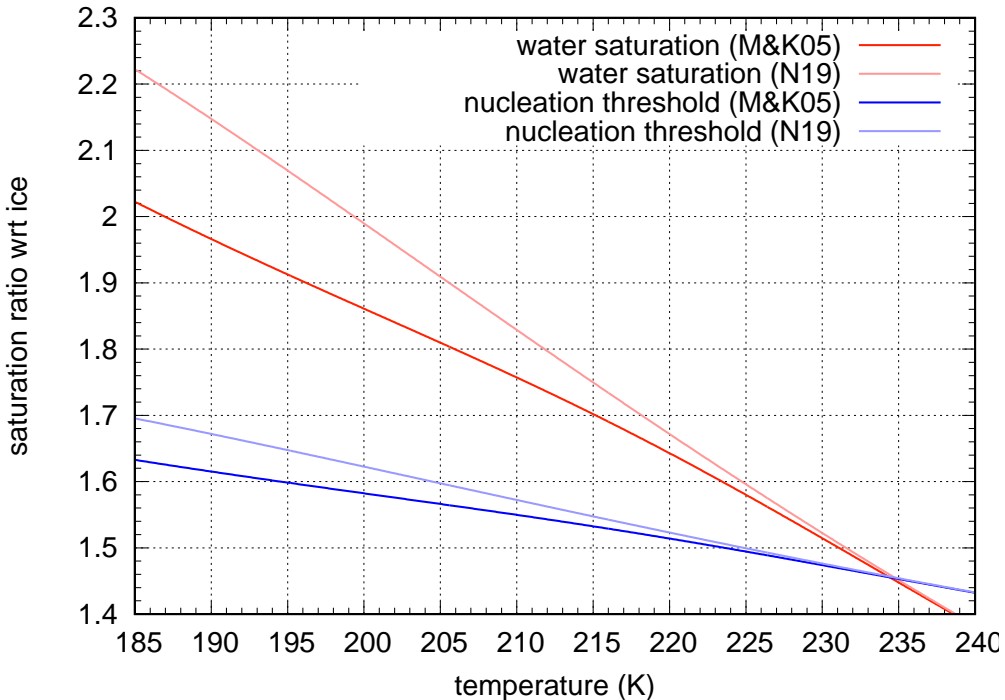

**Figure 9.** Water saturation ($S_i = \frac{p_{\text{liq}}}{p_{\text{ice}}}$) and nucleation threshold (for $J = 10^{16}\,\text{m}^{-3}\,\text{s}^{-1}$) for different formulations of saturation vapour pressure over super-cooled water, Murphy and Koop (2005) vs. Nachbar et al. (2019)

pressures over ice and liquid (i.e. the functions $a_w^i(T)$ and $\frac{1}{a_w^i(T)}$ behave differently: $a_w^i(T)$ is much closer to a quadratic curve as can be seen in the left panel of figure 10.

These new fits were used for the formulation of the approximated nucleation rate. Thus, we do not change the general
approach for approximating the nucleation rate etc., we only use a different representation of the function $a_w^i(T)$.

### 4.2 Numerical simulations of nucleation events

In figure 11 the results of the nucleation events using the new representation of the saturation vapour pressure due to Nachbar et al. (2019) are shown. As for former experiments, the ice crystal number concentration (left panel) and the maximum supersaturation values (right panel) are shown.

For the ice crystal number concentration, the impact of the new formulation of $p_{\text{liq}}$ is small; the relative deviation from the reference simulations using the original vapour pressure formulation by Murphy and Koop (2005) is always smaller than 15%. The deviation increases with decreasing temperature and is most prominent for lower vertical updrafts ($w < 1\,\text{m\,s}^{-1}$).

For the maximum saturation ratio the change as compared to the reference simulations is much more prominent. As can be seen in figure 9 the nucleation thresholds for a value of $J = 10^{16} m^{-3} s^{-1}$ are increasing with decreasing temperature





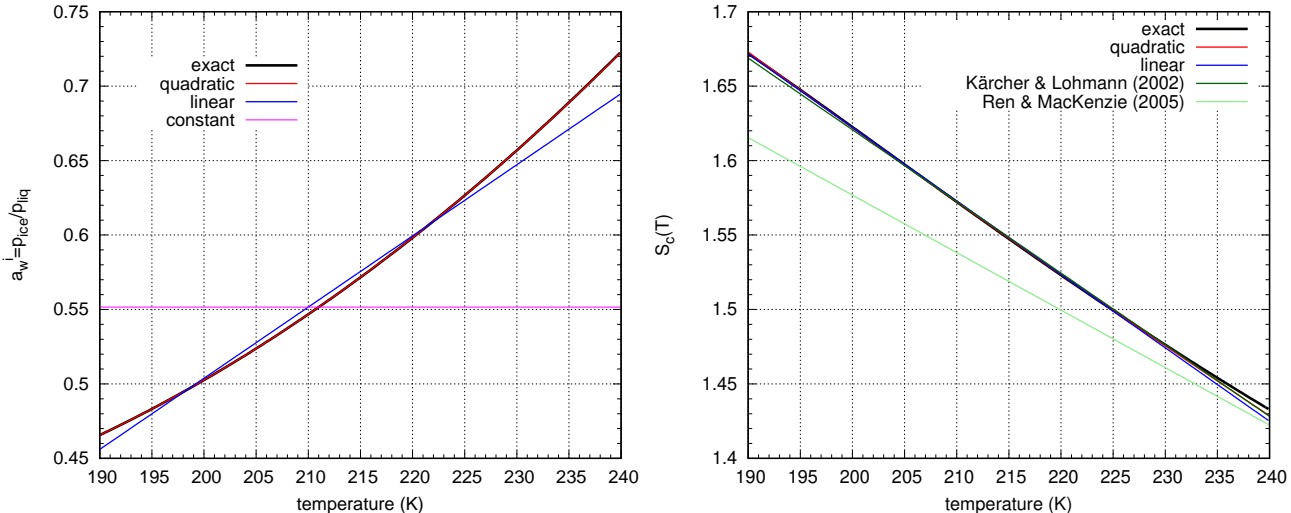

**Figure 10.** Left: Function $a_w^i(T) = \frac{p_{\text{ice}}(T)}{p_{\text{liq}}(T)}$ (black line) and polynomial approximations (red: quadratic, blue: linear, green: constant). Right: Nucleation threshold $S_c(T)$ (black line) and polynomial approximations(red: quadratic, blue: linear). Note that the former approximation by Kärcher and Lohmann (2002) (dark green) are now very close to the new formulation, whereas the fit by Ren and Mackenzie (2005) (turquoise) deviates significantly.

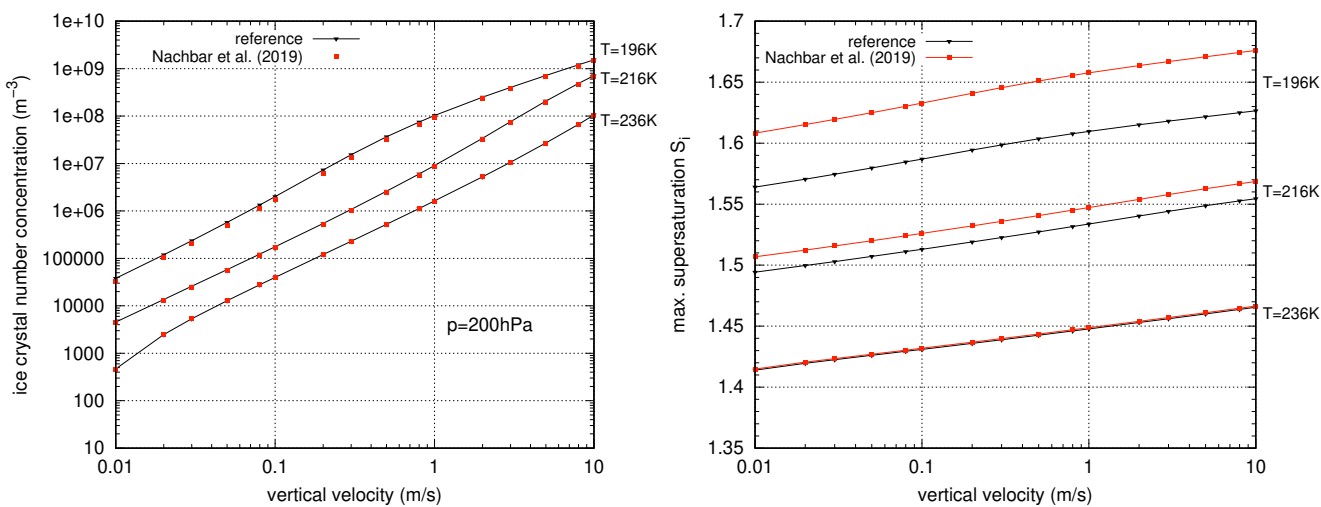

**Figure 11.** Impact of the formulation of the saturation vapour pressure by Nachbar et al. (2019) on the idealized nucleation events. Left: ice crystal number concentrations, right: maximum supersaturation values. The relative differences in number concentrations are always smaller than 15%

with a larger slope compared to the reference case. This behaviour can clearly be seen in the maximum supersaturation; for decreasing temperature the maximum supersaturation is increasing to higher values in comparison to the reference simulations.





The increase does not depend on the vertical velocities.

**Remark:** At the moment it is not clear, which thermodynamic hypothesis and thus which resulting approximation for the saturation vapour pressure over liquid water is physically correct. In particular, it is not clear if the formulation of Nachbar
et al. (2019) can be extrapolated to values $T < 200\,\text{K}$. Thus, we cannot recommend to use a certain formulation.

## 5 Another approach to formulate the nucleation rate

### 5.1 Direct fit to nucleation rate of pure water

In contrast to use the nucleation formulation by Koop et al. (2000) which is an excellent fit to laboratory data for freezing of solution droplets, we could also use the new fit for the freezing rate of pure water droplets (Koop and Murray, 2016). Here
we assume that at water saturation, the freezing of pure water droplets should behave as the freezing of solution droplets at super-cooled states. For deriving a new formulation based on the freezing rate as determined by Koop and Murray (2016), we reformulate the freezing rate depending on water activity $\Delta a_w$. Since the reformulation for a polynomial of high degree is quite complicated we just use a polynomial fit through the data as can be obtained from the original formulation $J_{\text{hom}}(T)$ at water saturation. For simplicity we use a quadratic polynomial, as described in appendix C. In figure 12 the original data and
the new fit is presented.

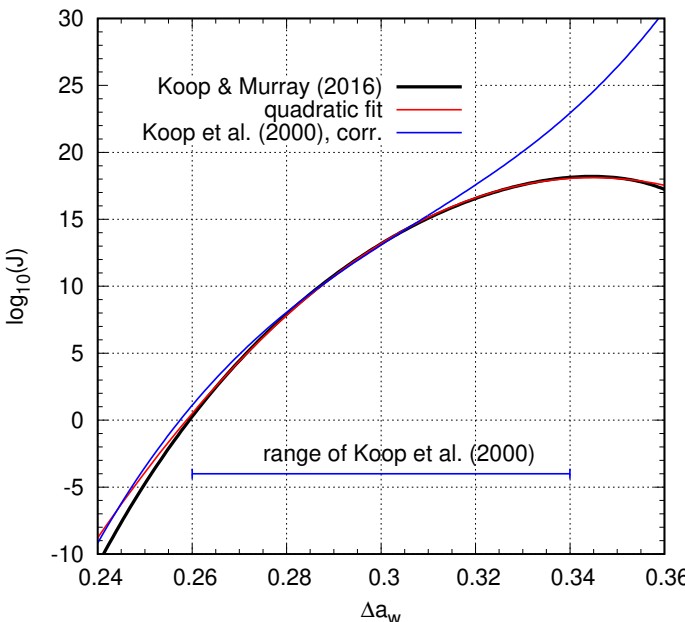

**Figure 12.** Freezing rate of water droplets (Koop and Murray, 2016, black), a polynomial fit (red), and the nucleation rate of solution droplets (Koop et al., 2000, corrected, blue), all depending on $\Delta a_w$.





Since the nucleation rate is now formulated in terms of $\Delta a_w$ it can be used for homogeneous freezing of solution droplets, assuming that the rate does not depend on other quantities than water activity. In comparison with the formulation by Koop et al. (2000) we find that the nucleation rate reaches a maximum at $\Delta a_w \sim 0.345$ and decreases afterwards. There is a significant deviation between the two nucleation rates ($J_{\text{K2000}} > J_{\text{KM2016}}$) for the range $\Delta a_w > 0.32$. Thus, we can expect that for cold temperatures and/or high upward motions there will be large deviations in the produced ice crystal number concentrations within nucleation events.

Since the formulation of homogeneous freezing of solution droplets (cf. Koop et al., 2000) relies on the shift of the melting curve by $\Delta a_w \sim 0.3$, the range of the parameterisation is restricted to the interval $0.26 \leq \Delta a_w \leq 0.34$. It is not clear, if the analogy works for values beyond $0.34$. In addition, some measurements for pure water droplets (Laksmono et al., 2015) show a kind of plateau for cold temperatures (which correspond to high values of $\Delta a_w$), which is not kept in the polynomial fit by Koop and Murray (2016). Thus, for higher values $\Delta a_w > 0.34$ we use the value $J_{\text{KM2016}}(\Delta a_w) = J_{\text{KM2016}}(0.34)$ to (a) mimick the plateau in the measurements, and (b) avod numerical issues in the simulations.

## 5.2 Numerical simulations of nucleation events

We investigate the impact of the newly proposed nucleation rate using numerical simulations as before. Since we have seen in section 4, that there might be an alternative way for formulating the saturation vapour pressure over liquid water (Nachbar et al., 2019), we carried out two different types of simulations for testing the impact of the nucleation rate, as derived in section 5.1: (1) Simulations using the standard formulation of $p_{\text{liq}}$ by Murphy and Koop (2005) and (2) simulations using the new formulation of $p_{\text{liq}}$ by Nachbar et al. (2019). The results of the simulations are shown in figure 13.

First we consider the ice crystal number concentrations (left panel). For low vertical updrafts, the values of $n_i$ are only slightly affected in case of using the adapted nucleation rate. For higher vertical velocities, there is a reduction in the produced ice crystal number concentrations; this reduction increases with increasing vertical updrafts. This effect can be explained as follows. The nucleation rates differ significantly for higher values $\Delta a_w \geq 0.31$, i.e. the slope of the adapted rate is (much) smaller than the original nucleation rate by Koop et al. (2000). For higher updrafts, the supersaturation reaches higher values, which is equivalent to higher values of $\Delta a_w$. Thus, the nucleation rates differ for these high updraft events, and less ice crystals are produced for using the adapted nucleation rate.

Generally, we can state that there is almost no difference in ice crystal number concentrations between the nucleation events using different formulations of the saturation pressure over liquid water, i.e. using the standard formulation by Murphy and Koop (2005) (red line in fig. 9) vs. the new formulation by Nachbar et al. (2019) (blue line in fig. 9), respectively.

Second, for the maximum supersaturation values, we see a similar behaviour as for $n_i$. For low vertical velocities there is almost no difference between the reference nucleation rate and the newly adapted rate. In case of using the saturation vapour pressure according to Nachbar et al. (2019), there is the shift in maximum supersaturation values depending on temperature, as we already saw in the simulations in section 4.

For higher updrafts ($w > 0.5\,\text{m}\,\text{s}^{-1}$), the maximum supersaturation values increase nonlinearly. For the coldest temperature ($T = 196\,\text{K}$) we note a dramatic increase up to very high values ($S_{i,\text{max}} \sim 1.8$). However, in all cases ice nucleation due





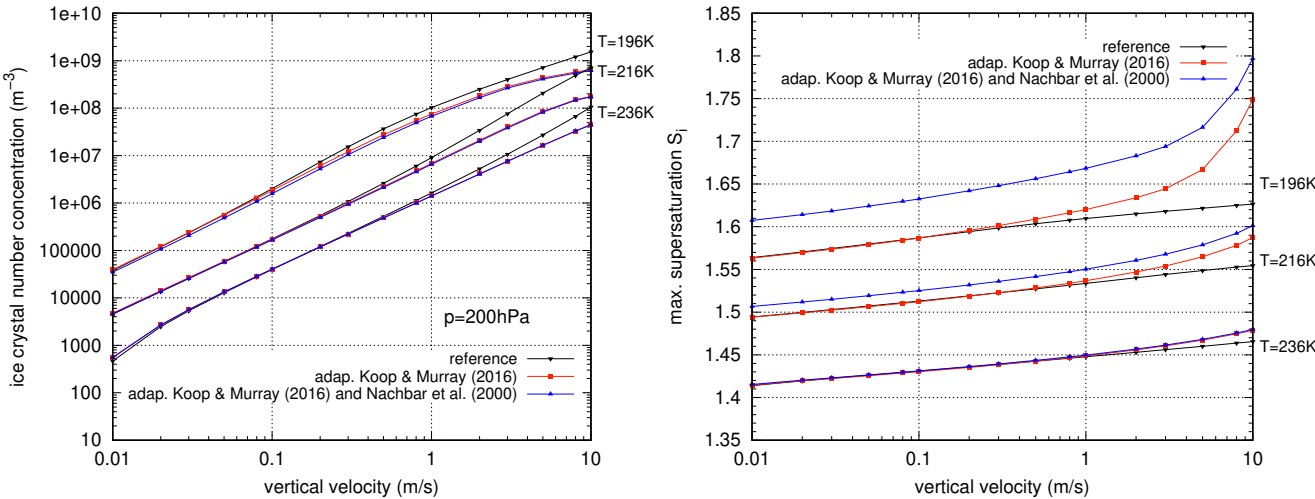

**Figure 13.** Impact of the direct formulation of the nucleation rate based on Koop and Murray (2016) on the idealized nucleation events. Black triangles and lines indicate the reference simulation, red squares and lines denote the use of the nucleation rate based on Koop and Murray (2016), and blue squares and lines represent the use of the nucleation rate based on Koop and Murray (2016) together with the saturation vapour pressure due to Nachbar et al. (2019). Left: ice crystal number concentrations, right: maximum supersaturation values.

595 to freezing of solution droplets is the correct prediction, since water saturation, and thus liquid origin ice formation, is not reached.

## 6 Thresholds of ice nucleation

For the evaluation of measurements of ice clouds, the possible range of supersaturation is often estimated using the so-called Koop-line, i.e. the supersaturation threshold $S_c(T)$ which corresponds to a nucleation rate value $J = 10^{16} m^{-3} s^{-1} = 10^{10} cm^{-3} s^{-1}$. In many investigations (see, e.g., Krämer et al., 2009) this function is used as an upper bound for possible values of $S_i$ inside and also outside of ice clouds. However, from our investigations in this study so far, we have to carefully consider two different aspects from a purely theoretical point of view:

1. The nucleation threshold assigned to the frequently used value $j_0 = 16$ is completely arbitrary chosen; there is no convincing physical justification for using this particular value; actually, in Koop et al. (2000) different values $J = 10^x m^{-3} s^{-1}$ with $x \in [1, 17]$ are used, but for testing the impact of droplet sizes, they used the value $x = j_0 = 16$. Nucleation of ice crystals is not a switching process, it occurs gradually and smooth, although the nucleation rates are very steep functions of the supersaturation. The size or strength of the nucleation event cannot be determined just by the maximum of the supersaturation; the amount of ice crystals as formed in the nucleation event is determined by the integral over the supersaturation curve (see, e.g., discussion in Dinh et al., 2016). Thus, it is possible to form many crystals in lower updrafts even if the high nucleation threshold is not reached. From our simulations we observe, that the





peak supersaturation for nucleation events depends crucially on the vertical velocity, i.e. on the temperature rate, which is prescribed during the event. This is quite obvious from the differential equation determining the change of $S_i$: The peak value is given by $\frac{\mathrm{d}S_i}{\mathrm{d}t} = 0$, i.e. when source and sink terms balance each other. Since the source includes the vertical velocity linearly, the dependence of the peak supersaturation on $w$ is obvious, although not linear.

615  2. As described above in section 4, it is still not clear which formulation of the saturation vapor pressure is physically correct. However, the use of the formulation by Nachbar et al. (2019) leads to a higher saturation vapour pressure and thus to a higher nucleation threshold, even for arbitrary values $x_0$ and its associated nucleation threshold $S_{cx0}(T)$.

Taking these two aspects into account, we can observe the following behaviour. In figure 14 (left panel) we compare the nucleation thresholds for the saturation vapour pressure according to Murphy and Koop (2005) for $j_0 = 10$ (red curve) and 620  $j_0 = 16$ (dark blue curve) with the range of peak supersaturations for vertical velocities $0.01\,\mathrm{m\,s^{-1}} \leq w \leq 2\,\mathrm{m\,s^{-1}}$ (black vertical bar) and the maximum value for a very unrealistic value $w = 10\,\mathrm{m\,s^{-1}}$ (black crosses) . For comparison, the well known Koop-line as fit and proposed by Kärcher and Lohmann (2002) is plotted (light blue curve). It is quite obvious, that for typical vertical velocity values the "classical" Koop-line is not reached, i.e. the peak supersaturation is below the threshold. Nevertheless, for strong cooling rates (very high vertical velocities), as are used in experiments in cloud chambers, high 625  supersaturations are reached, which still partly remain below the Koop-line. If we change the saturation vapour pressure to the formulation by Nachbar et al. (2019), the qualitative picture remains the same (right panel in fig 14): Even for high vertical updrafts the high nucleation rates are reached, for moderate and small updrafts, the peak supersaturation stays well below the classical nucleation threshold. However, the nucleation thresholds are generally shifted to higher values of supersaturation due to the different saturation vapour pressure formulation. It seems that these values fit better to the experiments in the AIDA 630  cloud chamber as reported in Baumgartner et al. (2022) and Schneider et al. (2021). This might be interpreted as a hint that the formulation by Nachbar et al. (2019) is the more appropriate formulation for the saturation vapour pressure. In any case, one has to consider the impact of the cooling rate on the peak supersaturation in a nucleation event. Therefore, the use of the "Koop-line" in the currently applied way is misleading and does not correspond to the actual physics of nucleation events. Note, that the threshold is used in some parameterisations of ice clouds in climate and numerical weather prediction models 635  (see, e.g., Kärcher et al., 2006; Köhler and Seifert, 2015).

Finally, we can also investigate the peak supersaturation values for the new empirical nucleation rate formulation, as derived in section 5.1. Generally, we see the same behaviour as for the reference simulations with a monotonic increase of peak supersaturation values with increasing vertical velocity (cf. figure 15). The use of the saturation vapour pressure formulation by Nachbar et al. (2019) additionally enhances the peak values as seen before. However, the peak values for cold temperatures 640  and very high vertical velocities are strongly enhanced in comparison with the reference simulations. Also these high values are still in line with the measurements in the AIDA chamber as reported by Baumgartner et al. (2022) and Schneider et al. (2021).


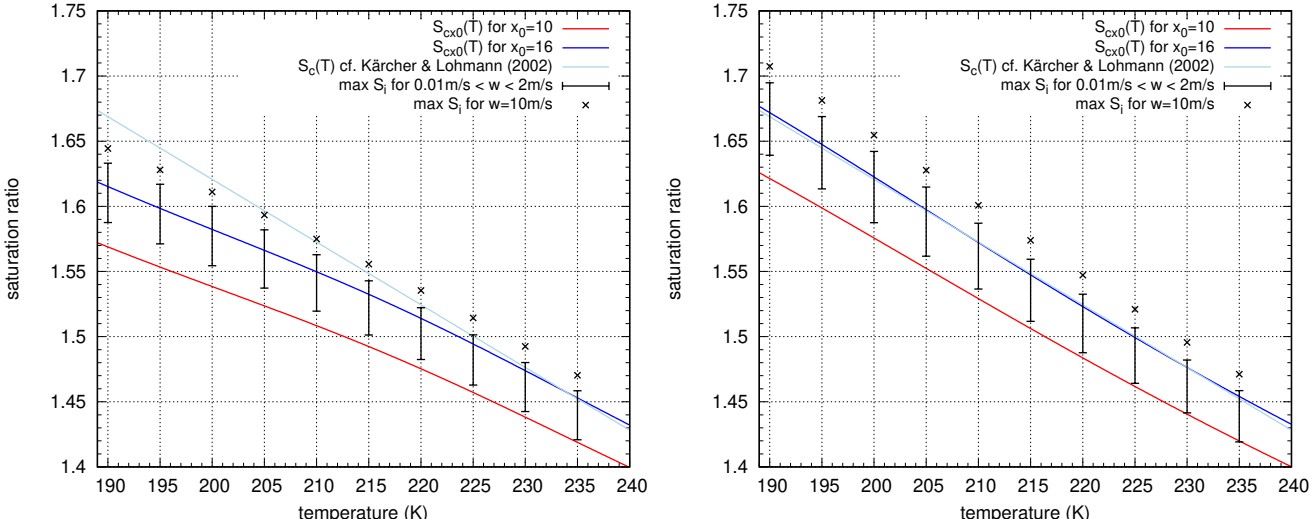

**Figure 14.** Comparison of nucleation thresholds (red curve: $x_0 = 10$, blue curve $x_0 = 16$) and the classical "Koop-line" (light blue curve). The black vertical bars indicate the range of peak supersaturation ratios within the nucleation events computed using vertical velocities ranging from $0.01\,\mathrm{m\,s^{-1}}$ to $2\,\mathrm{m\,s^{-1}}$. The black cross corresponds to the peak supersaturation ratio for the vertical velocity of $10\,\mathrm{m\,s^{-1}}$. Left panel: Curves based on the water activity using the saturation vapour pressure formulation by Murphy and Koop (2005); right panel: the same for the saturation vapour pressure formulation by Nachbar et al. (2019).

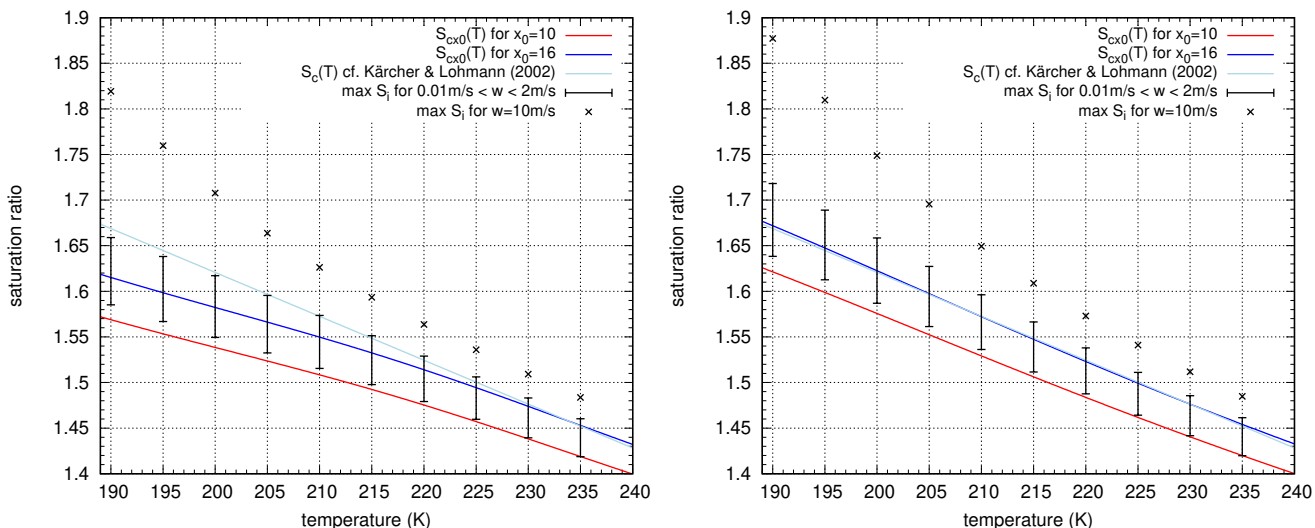

**Figure 15.** Same as in figure 14, but using the nucleation rate as empirically derived in section 5.1.



## 7 Summary and outlook

We have investigated the impact of the representation of nucleation rates and diffusional growth on idealized nucleation events,
as driven by a constant vertical updraft (i.e. a constant cooling rate). In a first step, we have investigated the original formu-
lation of the nucleation rate for homogeneous freezing of aqueous solution droplets in the formulation by Koop et al. (2000),
which is a well-accepted and verified formulation. For analytical purposes and simple model calculations, a less complicated
formulation is desired. We showed that a linear fit to the original formulation depending on the difference in water activity
$\Delta a_w = a_w - a_w^i$ is accurate enough to reproduce the ice crystal number concentrations quantitatively. Based on this lineariza-
tion approach, we derived a threshold formulation of the nucleation rate, which can be used for analytical investigations as
already presented in Baumgartner and Spichtinger (2019). Again, the new formulations are good enough to represent nucle-
ation events quantitatively as compared to the reference nucleation formulation.

Using the linear approximation as a starting point, we investigated the impact of different formulations on idealized nu-
cleation events, changing the two relevant parameters (slope and constant offset). These investigations led to the first major
results:

- The absolute values of the nucleation rate has only marginal impact on the resulting ice crystal number concentrations
  in a nucleation event. Even a scaling by up to six orders of magnitudes did not severely affect the resulting number
  concentrations. However, the maximum supersaturations changed, and the resulting deviations range up to few percent
  relative humidity. In addition, the time of nucleation onset is slightly shifted.

- The slope of the nucleation rate (or more precisely in the argument of the exponential function) has a much larger impact
  on the resulting nucleation event, and the ice crystal number concentration. Variations in the slope change the number
  concentrations in the nucleation events by up to a factor 2.5 (in both directions). Also, the maximum supersaturation is
  affected by a deviation of a few percent of relative humidity.

As a final conclusion of this part of our work, we can state that the shape of the nucleation rate is of high importance for the
representation of the nucleation process, whereas the absolute strength of the rate is almost negligible, if the values are high
enough. This shows that the nucleation process (homogeneous freezing of solution droplets) itself is a quite robust process,
thus the accurate formulation is maybe less critical as we thought. Also the amount of available solution droplets as controlled
by the background aerosol does not affect the nucleation events itself; it can be seen as a scaling factor of the nucleation rate,
in the same sense as in the sensitivity analysis of changing the absolute values of nucleation rates. As long as the amount of
aerosol particles is some orders of magnitude larger than the ice crystal number concentration as predicted for a nucleation
event, this does not play a role for the nucleation events.

We also investigated the impact of the recently published formulation of the water saturation pressure based on a thermody-
namic assumption of different phases of water in the very low temperature range (Nachbar et al., 2019). This new formulation
leads to changes in the function $a_w^i$, which directly affected the nucleation rate based on $\Delta a_w$. Following the derivations of
the threshold description, approximations could be found. The new resulting functions $a_w^i(T)$ and $S_c(T)$ can be accurately





approximated with polynomials of smaller degrees, as compared to the standard formulation. The new formulation of $p_{\text{liq}}$ only marginally changed the resulting ice crystal number concentrations. However, the impact on the maximum supersaturations increased with decreasing temperature up to few percent of relative humidity. Overall, the two different representations of the saturation vapour pressure over liquid water produced very similar, even almost identical, results. Thus, a decision about the validity of a certain formulation must be left to extensive experimental measurements.


In a more speculative part of the study we adapted the nucleation rate of homogeneous freezing of pure water droplets (Koop and Murray, 2016) as a new parameterisation for homogeneous freezing of aqueous solution droplets. This representation is quite similar for low values of $\Delta a_w$ to the original formulation by Koop et al. (2000) and its approximations. However, for very high water activities (i.e. high supersaturations as driven by large vertical updrafts), there is a significant deviation from the reference nucleation rate. Thus, for some cases in the parameter space (high updrafts and low temperatures) there is a significant deviation in the number concentrations, and, more obvious, in the maximum supersaturations, which reach almost water saturation in some cases. This approach showed that the shape of the nucleation rate is important for the resulting nucleation events; strong deviations of the shape from its reference affect the results of the nucleation event significantly. If this representation of the nucleation rate is close to the physics of ice nucleation remains open, and might be an objective for experimental investigations.



Finally, we investigated the commonly used threshold for homogeneous nucleation ("Koop-line") in the light of peak supersaturation values during nucleation events. This threshold corresponds to a nucleation rate of $10^{16}\,m^{-3}\,s^{-1}$, but is only rarely reached during nucleation events. Nucleation itself starts usually at much lower values of $S_i$ corresponding to lower values of the nucleation rate. The peak supersaturation during a nucleation event, characterised as an equilibrium between sources and sinks of supersaturation depends on temperature and vertical velocity. The peak supersaturation is a much more physical quantity to investigate the strength of a nucleation event. The peak supersaturation as diagnosed from the numerical simulations might be a more physical representation of ice nucleation in coarse resolution models in comparison to the frequently used nucleation threshold.


## Appendix A:  Model description - details


In this appendix, we present the details of the model as used for the numerical simulations of the nucleation events. Note, that we use the mathematical notation of logarithms, i.e. $\log$ denotes the natural logarithm (to base $e$).

**Background aerosol**

We assume for the aqueous solution droplets in the tropopause region a size distribution of lognormal type:

$$f_{\text{sol}}(r) = \frac{n_a}{\sqrt{2\pi}\log\sigma_r}\exp\left(-\frac{1}{2}\left(\frac{\log(r/r_{\text{sol}})}{\log\sigma_r}\right)^2\right)\frac{1}{r} \tag{A1}$$


with a modal radius $r_{\text{sol}} = 75 \cdot 10^{-9}\,\text{m}$, and a geometric standard deviation $\sigma_r = 1.5$. These values are adapted from the more complex model by Spichtinger and Gierens (2009), using the fact that the dry aerosol population, as used in Spichtinger and


Gierens (2009), has grown to larger sizes by water vapour uptake (i.e. assuming Köhler theory, see, e.g. Köhler, 1936). The mean volume of the solution droplets

$$V_d = V_{\text{sol}} = \frac{4}{3}\pi r_{\text{sol}}^3 \cdot c_{\text{sol}} \tag{A2}$$

with

$$c_{\text{sol}} = \exp\left(\frac{9}{2}\left(\log\sigma_r\right)^2\right) \tag{A3}$$

is calculated from the third moment of the lognormal distribution.

**Mass distribution for ice crystals**

For the ice crystals, we assume a mass distribution of lognormal type

$$f(m) = \frac{n_i}{\sqrt{2\pi}\log\sigma_m}\exp\left(-\frac{1}{2}\left(\frac{\log(m/m_m)}{\log\sigma_m}\right)\right)\frac{1}{m} \tag{A4}$$

with a parameter

$$r_0 = \exp\left((\log\sigma_m)^2\right), \quad \overline{m} = m_m\sqrt{r_0} = 3 \tag{A5}$$

representing the width of the distribution as described in Spichtinger and Gierens (2009). This distribution is used for the derivation of the rates in the ODE system for the mean quantities of ice mass and number concentration. The integration of

weighting functions of the type $m^k, k \in \mathbb{R}_+$ leads to general moments, which can be computed analytically:

$$\mu[m]_k := \int_0^\infty m^k f(m)\mathrm{d}m = n_i \cdot m_m^k \exp\left(\frac{1}{2}\left(k\log\sigma_m\right)^2\right)$$
$$= n_i \cdot \overline{m}^k r_0^{\frac{k(k-1)}{2}} \tag{A6}$$

Note, that for the averaged quantities we obtain $n_i = \mu[m]_0, q_i = \mu[m]_1$, respectively.

**Diffusion constant**

For the diffusion of water vapour in dry air, we use the following expression

$$D_v = D_{v0}\left(\frac{T}{T_0}\right)^{1.94}\left(\frac{p_0}{p}\right) \tag{A7}$$

which is an empirical fit to measurement data (Hall and Pruppacher, 1976). Note, that the valid temperature range is different in the book Pruppacher and Klett (2010) and in the original article Hall and Pruppacher (1976). For analytical investigations, a representation using a quadratic temperature dependence constitute a good approximation for a restricted temperature range. For the kinetic correction we use the function

$$f_D(r,a,b) = \frac{1}{\frac{r}{r+a} + \frac{b}{r}} = \frac{r^2 + ar}{r^2 + br + ab} \tag{A8}$$





whereas $r$ denotes the radius of the ice crystal (using a bulk density of ice $\rho_b = 0.81 \, \mathrm{kg \, m^{-3}}$), and the parameters are given by

$$a = \lambda \cdot C_{\mathrm{cunn}}, \quad b = \frac{4D_v}{\alpha_m \bar{c}_v} \tag{A9}$$

using the mean free path of water molecules in air $\lambda$ (acc. to Pruppacher and Klett, 2010), the Cunningham correction factor $C_{\mathrm{cunn}} = 0.7$, and the mean velocity of water molecules $\bar{c}_v$. We set the accomodation coefficient $\alpha_m = 0.5$ for comparison with former investigations (Kärcher and Lohmann, 2002); this value is also within the range as recommended in recent work by Skrotzki et al. (2013).

For representing the growth rates for the ensemble of ice crystals, by comparison with numerical integration we find that using a shifted mean mass $m_1 = c_1 \cdot \bar{m}$, $c_1 \approx 0.819$ in the kinetic correction function $f(r_1, a, b)$ is a good approximation.

**Howell factor**

Latent heat release due to phase changes during diffusional growth changes the surface temperature of the ice crystal. For taking this into account, we use the Howell factor

$$\begin{aligned}
G_v &= \left[ \left( \frac{L}{R_v T} - 1 \right) \frac{L}{T} \frac{D_v^*}{K_T^*} + \frac{R_v T}{p_{si}} \right]^{-1} \\
&\approx \left[ \left( \frac{L}{R_v T} - 1 \right) \frac{L}{T} \frac{D_v}{K_T} + \frac{R_v T}{p_{si}} \right]^{-1}.
\end{aligned} \tag{A10}$$

In the approximation, we neglect the kinetic corrections for diffusion coefficient $D_v$ and heat conductivity of air $K_T$.

**Capacity of ice crystals**

For ice crystals we assume columnar shape; thus the shape factor, or capacity, can be determined exactly using the electrostatic analogy (McDonald, 1963), using a prolate spheroid with semi axes $a, b$; the capacity can be analytically expressed by

$$C = \frac{L \epsilon'}{\log(\frac{1-\epsilon'}{1+\epsilon'})} \tag{A11}$$

using the eccentricity $\epsilon' = \sqrt{1 - \left( \frac{b}{a} \right)^2}$ and the length $L$ of the crystal. We find a very good approximation depending on the ice crystal mass

$$C(m) \approx a_1 \cdot m^{b_1} + a_2 \cdot m^{b_2} \tag{A12}$$

with constants

$$\begin{aligned}
&a_1 = 0.015755 \, \mathrm{m \, kg^{\frac{1}{b_1}}}, \quad b_1 = 0.3, \\
&a_2 = 0.33565 \, \mathrm{m \, kg^{\frac{1}{b_2}}}, \quad b_2 = 0.43.
\end{aligned} \tag{A13}$$

The representation of the capacity in the ice crystal ensemble is given by the integration, leading to general moments $\mu[m]_{b_i}$.





**Ventilation correction**

The empirical ventilation corrections usually depend on the use of two dimensionless numbers, i.e. the Schmidt number $N_{\mathrm{Sc}}$ and the Reynolds number $N_{\mathrm{Re}}$

$$N_{\mathrm{Sc}} = \frac{\mu}{D_v \rho}, N_{\mathrm{Re}} = \frac{\rho}{\mu} v_t L \tag{A14}$$

using the dynamic viscosity of air $\mu$ (e.g. Dixon, 2007). Thus, the size of the ice crystal $L$ is influencing the Reynolds number via the product $v_t(m)L$, using the terminal velocity $v_t$ for an ice crystal of mass $m$. The effect of ventilation, i.e. the additional

uptake of water vapour by the airflow around the particle crucially depends on the shape of the particles. For columnar shaped ice crystals, we adapt the empirical quadratic fit by Liu et al. (2003) to the simulation data (Ji and Wang, 1999) as follows

$$f_v = 1 + c_\chi \cdot \chi^2, \ \ c_\chi = 0.14856, \ \ \chi = N_{\mathrm{Sc}}^{\frac{1}{3}} N_{\mathrm{Re}}^{\frac{1}{2}} \tag{A15}$$

For the formulation of the terminal velocity of columnar shaped ice crystals, $v_t(m)$, we use the formulation by Spichtinger and Gierens (2009), including also the correction for temperature and pressure, respectively. For representing the ensemble of

ice crystals, by comparison with the numerical integration we find that using a shifted mean mass $m_2 = c_2 \cdot \overline{m}, c_2 = 1.5$ in the formulation of the Reynolds number leads to a very good agreement.

**Appendix B: Reference simulation results**

In this section we report on the results of the reference simulations, using the corrected formulation of the nucleation rate for super-cooled aqueous solution droplets by Koop et al. (2000). For evaluating the quality of the simplified model, we compare

the number concentration of ice crystals as obtained from standard nucleation events with results from literature, i.e. with a model using sophisticated Lagrangian particle physics (Kärcher and Lohmann, 2002) and a complex bulk physics scheme (Spichtinger and Gierens, 2009). In figure B1 the results are represented for the temperatures $T = 196, 216, 236\,\mathrm{K}$ at pressure $p = 200\,\mathrm{hPa}$, as prescribed in Kärcher and Lohmann (2002).

In comparison we see an overall good agreement of our simple model with the more sophisticated models (Kärcher and

Lohmann, 2002; Spichtinger and Gierens, 2009). However, we have to remark here that the deviation in the results for temperature $T = 236\,\mathrm{K}$ at low vertical velocities is the result of the neglegtance of the ventilation correction in the model by Kärcher and Lohmann (2002). In summary, our simplified approach compares very well with the results of the other studies.

In the study by Kärcher and Lohmann (2002) the impact of latent heat release on the diffusional growth is not considered. It is argued, that for cold temperature this effect is negligible. However, we found in our investigations, that this is only true

for temperatures well below $220\,\mathrm{K}$. A comparison at reference temperatures ($196/216/236\,\mathrm{K}$) shows, that there is an impact of latent heat leading to reduced ice crystal number concentrations due to an enhanced diffusional growth as compared to the values given in Kärcher and Lohmann (2002). For environmental conditions as above ($p = 200\,\mathrm{hPa}$), the reduction in the nucleated ice crystal number concentrations is about $20\%$ for $T = 236\,\mathrm{K}$ and about $5\%$ for $T = 216\,\mathrm{K}$, respectively. Thus, one





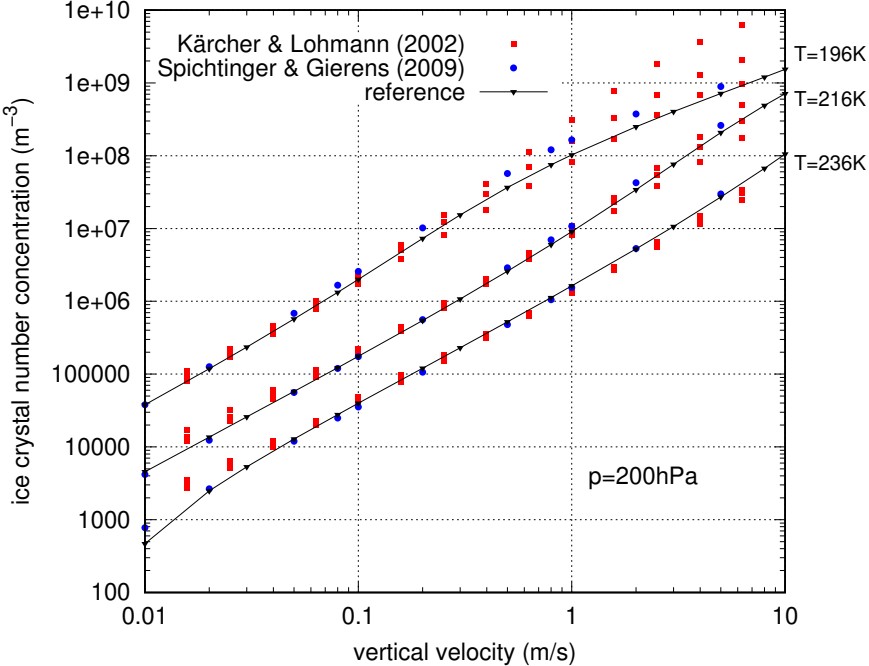

**Figure B1.** Comparison of ice crystal number concentrations as obtained for typical nucleation events from different models. Red squares: Particle model by Kärcher and Lohmann (2002), blue circles: complex two moment bulk scheme by Spichtinger and Gierens (2009), black line & triangles: simpler bulk model from this study, indicated as new reference

should be aware of that using parameterisations based on Kärcher and Lohmann (2002) might lead to moderately enhanced ice
crystal number concentrations in the warm temperature regime.

In figure B2 a typical nucleation event is shown. Here, two different nucleation parameterizations are used, the reference by Koop et al. (2000) (black line) and the linear fit (red line). There are small differences in the time evolution of the variables saturation ratio $S_i$ (left panel), number concentration $n_i$ (middle panel) and mean mass $m$ (right panel), but in general there is the same behaviour in both cases.

The source of supersaturation (i.e. cooling by vertical updraft and adiabatic expansion) leads to an increase in $S_i$ until nucleation starts at about $t_{start} \sim 40\,\mathrm{s}$, i.e. at very low values of the nucleation rate. $S_i$ is still increasing since the sink of depositional growth is not strong enough to reduce water vapour efficiently, thus the ice crystal number concentration is further increasing due to permanent ice nucleation. At the peak supersaturation, source and sink of supersaturation are balanced ($t_{peak} \sim$ 110 s); after this time, $S_i$ is decreasing due to the dominant growth term. The number concentration does not change much from
this time on but as long as the values of $S_i$ are large enough, still ice nucleation takes place. At about $t \sim 125\,\mathrm{s}$ the nucleation event is complete, no further nucleation takes place, since the nucleation rate is too small. Note that during the time interval $[t_{start}, t_{peak}]$ the mean mass $m$ is almost constant (this feature is more prominent in the linear fit case), whereas for $t > t_{peak}$ the mass increases. For $t < t_{peak}$ the nucleation is dominant, thus diffusional growth just compensates the number increase by



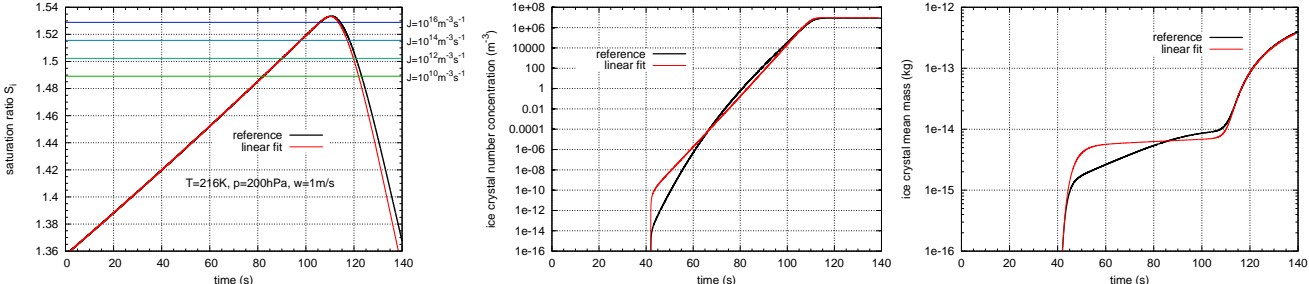

**Figure B2.** Representative example for a typical nucleation event for temperature $T = 216\,\mathrm{K}$ and pressure $p = 200\,\mathrm{hPa}$ with a forcing of $w = 1\,\mathrm{m\,s^{-1}}$. Red line: reference nucleation rate after Koop et al. (2000), black line: nucleation rate approximated by linear function as given in eq. (40).

mass, whereas afterwards crystal growth is dominant over nucleation. This feature was already seen in former investigations,

which leads to a model reduction for analytical investigations (Baumgartner and Spichtinger, 2019). The different nucleation parameterisations agree qualitatively for a nucleation event; however, the nonlinear reference rate leads to some variations. While for the linear fit case, the increase in $n_i$ is approximately an exponential growth $n_i(t) \sim \exp(\alpha t)$, and in turn the mean mass is almost constant in the relevant time interval, for the reference case the change deviates slightly from exponential growth.

Note, that the thresholds of constant nucleation rates in figure B2 (left panel) can be calculated from eq. (41) using the respective values for $j_0$ (i.e. $j_0 \in \{10, 12, 14, 16\}$) in the formulation of the supersaturation threshold.

**Appendix C: Simple fit for nucleation rate of pure water droplets**

In Koop and Murray (2016) a polynomial of degree 6 is used for fitting the experimental values of the nucleation rate for pure super-cooled water. Since polynomials of high degree are difficult to evaluate numerically, we present fits with polynomials of

lower degrees, which are still accurate in the relevant temperature range. The original formulation of the nucleation rate is

$$J_{\mathrm{hom}}(T) = 10^{p(x)}, \ \ p(x) = \sum_{i=0}^{n} c_i \cdot x^i, \ \ x = T - T_m. \tag{C1}$$

with a polynomial $p(x)$ of degree $n = 6$ using the melting temperature of pure water $T_m = 273.15\,\mathrm{K}$. The coefficients $c_i$ are reported in Koop and Murray (2016, table VII), where the nucleation rate is given in units $\mathrm{cm^{-3}\,s^{-1}}$. We reformulate the nucleation rate in SI units (i.e. $[J] = \mathrm{m^{-3}\,s^{-1}}$) by a factor of $10^6$ and approximate the logarithmic values $\log_{10}(J)$ by

polynomials of degree 2 and 4, respectively, i.e.

$$\begin{aligned} p_2(T) &= a_0 + a_1 \cdot T + a_2 \cdot T^2, \\ p_4(T) &= a_0 + a_1 \cdot T + a_2 \cdot T^2 + a_3 \cdot T^3 + a_4 \cdot T^4 \end{aligned} \tag{C2}$$





the coefficients are given in table C1. For this purpose we use a least square fit for the temperature range $225 \leq T \leq 245\,\mathrm{K}$, for which supercooled water droplets can still exist (see, e.g., figure 4 in Koop and Murray, 2016). In figure C1 (left panel) the approximations are shown in comparison with the original fit, while the ratio $r = \frac{p_i(T)}{p(x)}$ is shown in the right panel.

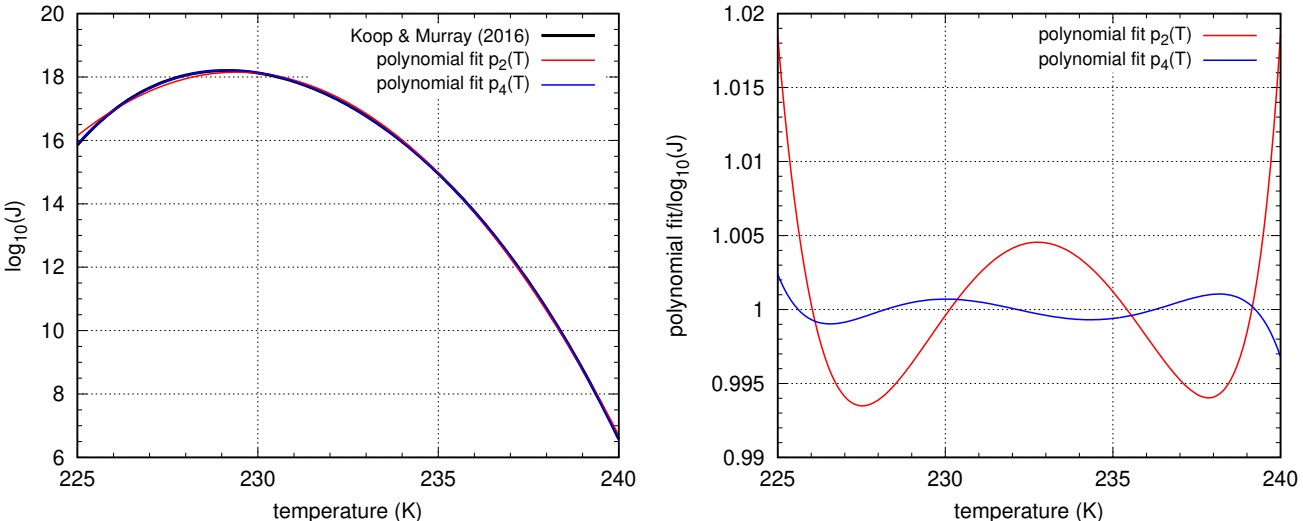

**Figure C1.** Polynomial fits of low degrees for the nucleation rate as given by Koop and Murray (2016). Left: Reference and fits $p_2(T)$, $p_4(T)$, right: ratio of reference and fits $p_2(T)$, $p_4(T)$

As can be seen the relative error for the polynomial fit $p_4(T)$ is less than 0.25%, while even for the quadratic fit $p_2(T)$ the error is smaller than 2%. For practical applications in the relevant temperature range $225 \leq T \leq 240\,\mathrm{K}$ the quadratic fit might be sufficient. If the original polynomial is used, a sophisticated evaluation of the polynomial is recommended (e.g. Horner scheme).

| fit | $a_0$ | $a_1$ | $a_2$ | $a_3$ | $a_4$ |
|---|---|---|---|---|---|
| $p_2(T)$ | $-5369.61$ | $46.96750$ | $-0.10236$ | – | – |
| $p_4(T)$ | $-848143.02$ | $14534.5767$ | $-93.481032$ | $0.26745460$ | $-0.0002872$ |

**Table C1.** Coefficients for the polynomial fits of the nucleation rate by Koop and Murray (2016) as given in equation (C2).

*Author contributions.* PM carried out calculations and approximations of the nucleation rates. PS run the numerical simulations. MB and PS
carried out the investigations using asymptotics. All authors were involved in the preparation of the manuscript and have read and approved
the final manuscript version.



*Competing interests.* The authors declare that they have no conflict of interest.

*Acknowledgements.* Patrik Marschalik and Peter Spichtinger acknowledge support by the German Bundesministerium für Bildung und Forschung (BMBF) within the HD(CP)$^2$ initiative, project S4 (01LK1216A). Manuel Baumgartner acknowledges support by the Deutsche Forschungsgemeinschaft (DFG) within the Transregional Collaborative Research Centre TRR165 Waves to Weather, (www.wavestoweather.de), project Z2. Peter Spichtinger acknowledges support by the DFG within the Transregional Collaborative Research Centre TRR301 TPChange, (tpchange.de), project B7. We thank Martina Krämer for fruitful discussions during her stay as a Gutenberg Research College fellow at JGU Mainz.



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
