# Peer review of "Impact of formulations of the homogeneous nucleation rate on ice nucleation events in cirrus"

_Atmospheric Chemistry and Physics, 2022_

## Author Comment (AC1)

**Response to reviews**

P. Spichtinger, P. Marschalik, M. Baumgartner,

**1 General response**

First of all, we thank both reviewers and Bernd Kärcher for their helpful comments and suggestions which lead to a significantly improved manuscript. This document contains our responses to the reviewer comments. We list the reviewer comments (in blue) below together with our responses.

We have rewritten the introduction in order to clarify our motivation (see also response to reviewer#1). For a better overview, we shifted the part on the nucleation rate into a new section (Empirical fit of the nucleation rate). We also have rewritten the model section and the section on the new formulation of the nucleation rate adopting the results by Koop and Murray (2016).

**2 Response to reviewer #1**

"In this study we will investigate the impact of the formulation of the nucleation rate on the resulting ice crystal number concentrations." I miss the motivation for this re-assessment. Has it turned out recently that the usual formulation (using Koop et al. 2000 essentially) is wrong? The authors mention that the old Mishima-Stanley theory is now ruled out and that two alternative water theories have survived which give similar results for upper-tropospheric conditions. It is not clear to me, however, how this fact impacts on the Koop-etal 2000 findings and their formulation of the nucleation rate coefficient. The implications of this finding is not given in the paper, and in fact, these theories are not mentioned again. It is thus unclear why this is mentioned at all.

We have rewritten the introduction in order to make clear our motivation. Actually, the Koop formulation of the nucleation rate is not wrong (but maybe also not correct), but it is merely a fit to experimental details. The choice of the cubic polynomial was motivated by the formulation of the nucleation rate for pure supercooled water by Pruppacher (1995). In essence, this fit tries to mimick the singularity in the rates as predicted/suggested by the Mishima-Stanley theory (leading to a cubic polynomial) but this theory is ruled out since quite a while. Moreover, new measurements for freezing supercooled water droplets are available, we can now ask ourselves if nucleation rates for aqueous solution droplets can be formulated differently, or how uncertainties in the actual fit propagate to the ice crystal number concentration, resulting from nucleation events. Finally, we would like to have a simple and robust formulation of the nucleation rate for the use in numerical models and in particular for analytical investigations.
We have added these points for the motivation in the introduction.

"However, the representation of these processes contains still uncertain parameters or even the (mathematical) formulation of the processes remain uncertain." Perhaps this is the point where the authors should become more concrete. What are the uncertain parameters and in which way is the mathematical formulation uncertain.

Line 75: It would be good to list the mentioned parameters.

Since the whole fitting procedure assumes a cubic polynomial on the basis of a theory, which is now obsolete, one can questioning the whole representation of the nucleation rate. Since we have rewritten the whole introduction/motivation, this vague statement has been deleted.

Lines 105 ff: It is not easy to understand the statement "constant temperature and pressure is assumed" when you just before present the T and p tendency equations. My understanding is this: you do not need p and T themselves, all you need is the changing supersaturation which in turn can be prescribed, such that the proper causes for changing S, namely changing p and T, need not to be explicit. If my interpretation is correct, I suggest that the authors change their explanations accordingly. In this case, why do we still need p_env and T_env, is this because there are p and T dependent parameters in the equations (e.g. diffusion)?

We followed the recommendation of the reviewer. We prescribe just the saturation ratio and use the constant temperature and pressure to evaluate other physical parameters (such as the diffusion coefficient) in the model.

Line 120: You may add that such a high value is mainly chosen in order to let the nucleation process run without consumption of the available aerosol droplets, which would complicate the interpretation of the results.

We followed the recommendation of the reviewer and added a statement about the high values.

Eq. 20: As aw and Delta aw have a special value for water saturation, say Delta aw* I would prefer to have the equation in this form J_sol(Delta aw*[T])=J_pure liq(T).

We changed the notation accordingly.

Line 164: Can we understand why this equality fails?

It is not completely clear, why the equation fails, i.e. why the nucleation rate for pure water droplets as derived by Koop and Murray does not agree with the nucleation rate of solution droplets. We would guess that the different derivations of the fits and the different measurements might play a role. However, the reason is not clear, a detailed investigation of the underlying experiments is beyond the scope of this study.

Line 168: strictly speaking, the log of the nucleation rate is shifted, which means that the nucleation rate itself is to be multiplied by a constant (obviously a constant far away from unity).

We followed the recommendation of the reviewer

Figure 1 caption: Not completely clear. I think the blue line is for solution droplets with infinite dissolution, that is pure water droplets. This should be clear in the figure caption.

We added some words in the figure caption

Line 176: I still don't understand this. Probably I did not follow the most recent literature, but who (reference) has found that the Koop etal 2000 parameterisation is wrong? Can the results of Koop and Murray for pure water be used as a proof for the falsehood of the older formulation when the latter is applied to solution droplets? In order to accept the shifted Koop 2000 line as a reference, one needs to know what is wrong with the unshifted original line. As far as I remember, this was based on measurements as well. Have there been indications that these measurements were incorrect? Later you demonstrate that differences between several formulations and your reference formulation are as small as typical measurement uncertainties and that it can therefore not be decided, which formulation is "correct" (I would rather say, which formulation is most appropriate, since "correct" implies that you know the truth.) So it seems, the motivation for your exercise is not that the original formulation is wrong, but rather that you seek for something quite simple that can be used in analytical studies. If so, this should be clearly stated in the beginning.

There are no recent measurements indicating that the formulation by Koop et al. (2000) is correct or wrong, although the disagreement with the recent formulation of the nucleation rate for pure water as based on new measurements by Koop and Murray (2016) can be stated. This recent formulation together with the reassessment of the water theories lead us to the question how sensitive nucleation events are in terms of changes in the current formulations. So, the reviewer is right in formulating our main motivation, and we have put this into our new text in the introduction. We also avoided the term "correct" in the following, because we do not know the truth.

Line 202: replace "explanation" by "interpretation".

We changed the word and added some short description.

Line 211: Isn't the degree of p(x) fixed by your ansatz in Eq. 25?

Yes, this is correct, we added some words for clarifying.

Line 286: not clear, what the vertical axis here represents.

We changed the text to "The threshold function is just shifted by the value $\frac{x_0 - j_0}{A(T)}$, i.e. the type of the threshold function remains the same."

Section 3.5.2: I don't get the point here. Before you have made a best fit, but what is interesting in showing results obtained with worse fits? I think, this section can be shortend without much loss to the reader.

The original parameterisation of the nucleation rate by Koop et al. (2000) is based on a cubic polynomial, which changes its slope in the range $221 \leq b \leq 453$, as we found out in section 3.3. (new section 4.3.) using Taylor approximations to the cubic polynomial. Thus, these values at least might be relevant for a choice of a linear fit. This fact acts as a motivation to investigate the impact of changes in the slope, which can be easily carried out in our linear formulation. In addition, it is generally interesting if the absolute value of the nucleation rate does really impact the nucleation events. This is the motivation for choosing a "worse" fit, which is actually a sensitivity study.

We added some explaining text at the beginning of the section.

Section 3.6: I must admit, that I cannot follow in this section. This kind of mathematics is not in my repertoire. I fear that other readers can have problems as well with this section. I suggest that a short paragraph be added, either here or as an appendix, where this technique is shortly explained.

We followed the suggestion of the reviewer and added an appendix explaining the main idea of asymptotic analysis, which is actually perturbation theory.

Section 5: Please make clearer what is new here to the developments in Section 3. In 3, the reference nucleation rate was the Koop et al 2000 formulation, but shifted such that the Koop and Murray results for pure water were matched in a certain T-range. It is not quite clear to me what the difference of this to the new fit is.

In section 3 (new section 4) we just rescaled the whole nucleation rate as formulated by Koop et al. (2000) in order to match the new parameterisation by Koop and Murray (2016) in a certain temperature interval. As discussed, from theory we would expect that both curves match over the whole temperature range. In this section 5 (new section 6) we take a different point of view assuming that we can just adopt the Koop and Murray formulation for the nucleation rate of aqueous solution droplets, providing an exact match of both curves by definition. Thus, the newly formulated nucleation rate of aqueous solution droplets based on Koop and Murray (2016) shows strong deviations for high values of $\Delta a_w \geq 0.31$; instead of a monotonic increase, there is a maximum at about $\Delta a_w \sim 0.34$. This deviation leads to a different behaviour in nucleation events.

We have rewritten section 5 (new section 6) and have also included some explanation as above.

Line 572: "avoid"

Changed

Line 634: This is an important remark. Can you give any recommendation how this formulation should be changed in such models to become more realistic?

A simple first step would be to change the threshold function for switching on nucleation during a coarse model time step to a function depending on temperature and vertical velocity. For this purpose one could calculate the maximum supersaturation values for a dense sampling of the parameter space $185 \leq T \leq 240$ K and $0.001 \leq w \leq 10$ m/s and use a 2D fit for a threshold $S(T, w)$. However, even this approach is underestimating the onset of nucleation, which usually takes place at lower values $S_i \leq S(T, w)$.

**3 Response to reviewer #2**

1) Consistency between liquid saturation pressure and nucleation rate: As far as I understand, for consistency with the Koop et al. (2000) method, each change in the saturation formula (and implied change in ice water activity) requires updating the fit to the homogeneous freezing rate of pure water. Was this carried out when switching to Nachbar's formula in section 4 ? It should be specified if yes, done if not.

Yes, we calculate the values of $a_w^i(T) = \frac{p_{\mathrm{ice}}(T)}{p_{\mathrm{liq}}(T)}$ or appropriate approximations using the respective formulation of the saturation vapour pressure. This is now indicated in the text.

Related issue: the authors have already published on the impact of Nachbar's formulation in Baumgartner et al. (2022). Differences should be highlighted. If there are none, the whole section 4 (including Figures 10 and 11) can be removed and summarized in just a couple of sentences (in particular since the effects are small).

In Baumgartner et al. (2022) the focus was on a different topic, i.e. more on size effects and formulations of the water activity from a physical chemistry viewpoint. Although some of our results are similar to the results in the former study, we want to keep them in order to have a consistent picture, in particular in combination with new results (e.g., on maximum supersaturations). Nevertheless, we have shorten the text and deleted figure 8.

2) Clarity and organization: I would suggest the authors start by presenting the uncertainties regarding the nucleation rate and the new formulations before introducing their approximations and finally analyze the impact of the updated formulas. This would make the motivation clearer. I also think the manuscript would benefit from shortening. Many equations and comments are redundant (see below).

We have rewritten the introduction in order to make clear our motivation for this study. Actually, the starting point is the mismatch of the nucleation rates for pure water and solution droplets, as well as the lack of empirical basis for the cubic fit. We also put the information about the nucleation rates into a new section. Finally, we again went through the manuscript to shorten the text and to delete some redundant equations and descriptions.

3) Suitability of the model to study details of ice nucleation: in a comment to this paper, Bernd Kärcher expressed skepticism regarding the ability of a bulk model to study ice nucleation. For such a focused study, expected limitations of the modeling approach should be discussed.

It was demonstrated in many studies that bulk models are well suited for describing the major properties of ice nucleation. Even the impact of heterogeneous ice nuclei can be described quite well. The relation between vertical velocity and produced ice crystal number concentrations as derived with complex bin/particle models can be reproduced with bulk models in a very robust way. Therefore, we can be quite sure that this feature is not observed by accident.

There are many examples in literature for using bulk schemes and especially investigating nucleation events in comparison with observations, in parcel models (e.g., Baumgartner et al., 2022) and also in LES models for different scenarios (e.g., gravity waves in Joos et al., 2009; convective cells in Spichtinger, 2014). In all these studies the overall agreement with measurements was quite good.

The question of the suitability of a certain model must be reformulated in terms of the ability to represent main features on distinct scales. To our knowledge, there is no rigorous treatment of this question. Nevertheless, based on the examples above we can carefully conclude that bulk models might be well suited on scales on few 10s of meters up to larger scales.

In addition, we added a short text in the introduction and the summary that all the results of this study are only valid in the bulk-view. If one is interested in the formation details of a single ice crystal (or only a very small number of particles) then all the details in the nucleation rate might be of high importance (in contrast to our results for the bulk).

4) What is the reason for the simple adjustment to the Koop et al (2000) formula in Sect. 3.1? This seems to create a 'third' mixed category between the former and updated formula. I would recommend restricting the study to those two cases.

As described in the new section, the nucleation rate of pure water droplets and the nucleation rate of solution droplets should agree for water saturation. Obviously, this is not the case. However, in the relevant temperature interval between 235 and 240 K both rates agree qualitatively, but there is a small offset resulting in a factor of $\sim 10^{1.5} \approx 30$ in the nucleation rate. Thus, we adopted the original formulation by Koop et al. (2000) to create a best match. It is not a formulation in between, but an improvement of the original formulation by Koop et al. (2000), which then serves as a reference for approximations of the nucleation rate.

5) I had difficulties following some sections, in particular 3.6. At other places, the text sometimes falls into tautology: in section 3.4, the authors arrive at Eq. (45) which as far as I understand is exactly the same as Eq. (43) and (42) if we consider that Sc also depends on j0. Eq.(42) is repeated only a few lines below as Eq.(43) on page 11.

We went through the text in order to streamline the manuscript, several equations were deleted. In section 3.6., we added two examples explaining the results of the analysis. In addition, we added an Appendix for giving an idea ofasymptotics/perturbation analysis.

6) There are some ambiguities in the notations, for instance pressure and the polynomial are both represented by the same symbol (p) . I suggest clarifying and adding a table with a list of symbols.

For pressure we still use the letter $p$ but we added an index for all polynomials that corrensponds to the degree of the polynomial. After the revision of our notation, we are confident that all ambiguities are removed. Moreover, apart from the notation of standard physical quantities the mathematical symbols have no universal meaning, hence we refrain from adding a separate list of symbols.

7) I am not entirely convinced about section 6. The relevant quantity for atmospheric modeling is the ice crystal number density, not really the threshold which is mostly specific to chamber experiments. Moreover, the 'Koop-line' depends on the aerosols size distribution and is not supposed to represent the maximum supersaturation reached, rather an approximate ice onset.

In many coarse grid models, the "Koop-line" is used as a threshold for switching on ice nucleation (e.g., Kärcher and Lohmann, 2002). The Koop-line itself depends not on the aerosol size distribution, but is derived for a certain volume of a solution droplet ($0.25\mu$m) and a certain value of the nucleation rate $J = 10^{1}0\text{cm}^{-3}\,\text{s}^{-1}$, as described in Kärcher and Lohmann (2002, their eq. (4)). In our framework, this is just the supersaturation threshold as derived for a certain nucleation rate value $x_0$, as described in the manuscript. We agree that the maximum supersaturation is not describing the onset of the nucleation, but this is also not the case for the Koop-line. An adaptation including vertical velocity as described in the manuscript (we added some text) might lead to a more realistic treatment of the onset of nucleation. We added some more words on this issue in the manuscript.

In any case, the maximum supersaturation would be important for the evaluation of measurements, since the Koop-line is often not reached for low vertical updrafts, as shown in the respective section.

8) Diffusion growth: it seems that it was originally intended that the paper also treats the sensitivity to diffusion growth (the summary line 644 still mentions it). While I agree this is beyond the scope of the study, could the authors comment in the text on whether the sensitivity to nucleation rate formulation they characterize holds for different growth parameters? Also, it would make sense in this case to have Part 1-Part 2 papers.

Actually, we have investigated the impact of diffusional growth on nucleation events, too. We decided to remove it from the text. We went through the manuscript in order to delete the occasionally appearing statements on diffusional growth.

specific comments

- Title: should mention homogeneous nucleation or homogeneous freezing since it is the only nucleation pathway considered in the paper.

  We changed the title to

"Impact of formulations of the homogeneous nucleation rate on ice nucleation events in cirrus"

following the recommendation of the reviewer.

- P4 Model description should state that the model has two moments.

We added some text to clarify this.

- Line 111 and 132: There is an inconsistency, it is first stated that latent heat release is neglected but this term is still included later.

We have rewritten the model description, this inconsistency is now removed.

- line 140-141: Same as above, the first term comes from considering latent heat.

We have rewritten the model description, this inconsistency is now removed.

- Line 164-165: This disagreement at temperatures above 235 K is indeed surprising. Have you confirmed by comparing with the original Pruppacher 1995 data used by Koop et al. (2000) ?

Koop et al. (2000) did not use the data by Pruppacher (1995). However, they used the cubic polynomial ansatz for consistency with Pruppacher (1995). The direct comparison was not carried out before, as far as we know.

- line 181: missing index n

We added an index $n$ for all polynomials in the manuscript

- line 258: eq. 41 is the same as eq. 31

This is not completely correct, they refer to different constants of the linear approximation, therefore we would like to keep them.

- line 340, 'most': not all ?

Unfortunately not "all", because there is an outlier for T=236K and w=0.01m/s. We changed it to "almost all".

- line 701: ISO convention for natural logarithm is ln , not log

This is correct. However, since our investigation is based on a mathematical framework and in most computer languages "log" is also the logarithm to base e, we prefer to stay with "log". We also do not want to change $\log_{10}$ to "lg", which would be ISO convention.

- Line 747-748: Please check formula A11 and please correct if needed. Also define L.

We corrected formula A11, this was a typo. We additionally defined $L$ as the length of the columnar shaped ice crystal.

- Line 751: for which value of the eccentricity ?

The eccentricity depends on the length of the crystal. Actually, the aspect ratio of the crystal changes if it growths (see Spichtinger and Gierens, 2009, eq. (17)), thus the capacity changes. in order to avoid piece-wise functions as in Spichtinger and Gierens (2009), we use the new approximation eq. (A12).

**4 Response to Bernd Kärcher**

Ad I  The first sentence in the paper is a strong statement demanding justification. Clearly, homogeneous freezing of solution droplets is a fundamental atmospheric ice formation process that operates when ice-nucleating particles are either absent or present in very low concentrations. Any assertion whether and in which circumstances this pathway dominates 'ice formation in cold temperature regimes' (an active area of research) needs to be supported by appropriate references. What about the role of mineral dust in cirrus cloud formation [Froyd et al., 2022], with dust particles long known to be very efficient heterogeneous ice-nucleating particles?

We changed the abstract in such a way that it is clear that we only investigate homogeneous freezing of solution droplets as an important ice formation mechanism.

Ad II  I don't quite understand the categorization in-situ vs liquid origin formation. In the absence of cloud water droplets, freezing agents include solution droplets, and also in this case, it is the liquid water that freezes. Plus, situations where cloud droplets freeze are also in-situ events. Does liquid origin formation distinguish between ice formation in convective detrainment zones and in conveyeor belts, in which either solution and cloud droplets may freeze?

The terms in situ formation vs. liquid origin formation are now well-established terms since their introduction in papers by Krämer et al. (2016) and Wernli et al. (2016). In particular in the study by Wernli et al. (2016) these terms are carefully defined. There is also a short explanation in the manuscript.

Ad III  Why would the nucleation event be idealized? I think it is the modeling that is idealized.

The scenario of a constant updraft/cooling rate is idealized in the sense that in the real atmosphere updrafts are not constant over the whole time period that is considered.

Which of the microphysics schemes referred to / used in the manuscript is 'state-of-the art' and how is this attribute defined?

Our scheme can be seen as state-of-the-art microphysics scheme for bulk models, since it is using a reference formulation for the nucleation rate and diffusional growth; the formulations are physically and mathematically consistent.

The numerical scheme used in Kärcher & Lohmann (2002) does not employ Lagrangian particle physics, as claimed in line 771.

Thank you for the comment about the scheme in Kärcher and Lohmann (2002), we have corrected this statement.

In that context: could you please motivate / explain more clearly why "... details of the nucleation rate are less important for simulating ice nucleation in bulk models ..."(abstract line 6) only relates to bulk models?

We have rewritten the abstract and have deleted the term "bulk models", since this was misleading. Our investigation is based on bulk models, however, it is quite obvious that changes in the nucleation rates would have a similarly small impact for more sophisticated, e.g. particle based, models.

Ad IV  Latent heat release enhancing temperatures during homogeneous freezing of cloud water droplets roll off around 234K depending on the CCN spectrum [Kärcher, 2017]. Here, the authors assert to find moderate (significant?) reductions of homogeneously nucleated ice number concentrations down to much colder temperatures due to latent heat release, this should be demonstrated by numerical simulations. Please analyse the individual diabatic contributions due to water phase changes to the temperature budget with the dry adiabatic tendency at 220K and show a quantitative comparison of homogeneously nucleated ice crystal number concentrations between simulations across a range of updraft speeds with and without including effects of latent release.

We deleted this paragraph in the manuscript, since it is not relevant for our investigations. It is for sure beyond the scope of the study to provide a complete analysis of diabatic contributions to the respective terms in the formulation.

Ad V  The point that homogeneous freezing thresholds depend on cooling rate and droplet size is known [Kärcher et al., 2022]. I am aware of the difficulty to oversee the onslaught of manuscripts in the scientific literature. In Kärcher et al. [2017], we have analysed the microscale characteristics of homogeneous freezing events in detail. Both articles are relevant for the present manuscript and therefore the authors may want to update their reference list.

It is completely obvious from theory that the maximum supersaturation is depending on the prescribed vertical updraft, as already stated by Korolev and Mazin (2005). However, as we show in our study, the maximum value depends on details in the nucleation rate, which cannot be determined analytically, but only in the simulations.

I express strong doubts that a w-dependence of the freezing supersaturation threshold might explain low temperature cloud chamber data; if the authors have indications for this to be a valid explanation, it should explicitly be included in the study, focussing on the region T<205 K. Also, the treatment of the whole issue in the manuscript is somewhat confusing, as the authors admit in line 631 that "This might be interpreted as a hint that the formulation by Nachbar et al. (2019) is the more appropriate formulation for the saturation vapour pressure.", i.e., invoking a low-T correction of the currently used saturation vapor pressure as the main cause of high freezing thresholds and not deviations from a fixed freezing threshold in parameterizations of cirrus cloud formation.

It was not claimed that a $w$-dependent threshold explains the measurements in the chamber experiments. It is just that fact that the maximum supersaturation for nucleation events depends on vertical updrafts, thus leading to higher values than the Koop-line. The high values above the Koop-line can just be better explained on the basis of the maximum supersaturations. Comparing measured values of $S_i$ with different approaches of the nucleation rate leads us to the interpretation that some of them agree better - in the same way as it is done in Baumgartner et al. (2022), but with a different focus.

Also, it remains unclear why this suggests a reformulation of ice nucleation schemes used in coarse models. In cirrus, the dependence of nucleated ice numbers on the threshold supersaturation is relatively weak (less than linear, see Kärcher & Lohmann, 2002), let alone the logarithmic dependences on the cooling rate, the liquid water volume in solution droplets, and the total droplet number concentration. These dependencies are consistent with / well explained by the self-terminating homogeneous freezing-relaxtion mechanism;recently your team recognized based on numerical work that an increased nucleation threshold has little impact on ice crystal numbers (Baumgartner et al., 2022).

The comparison with measurements do not suggest to change the parameterisations. However, from the investigation of the maximum supersaturation in comparison with the nucleation thresholds for a given rate ($j_0 = 10/12/14/16$) it turns out that for low velocities the Koop-line (as a usual threshold for triggering ice nucleation) is not reached. Thus, it might be worth to think about a possible adjustment of the parameterisation. The exact treatment of such a possible parameterisation on basis of variables as, e.g., temperature and vertical updraft, is beyond the scope of this study. However, to answer the first reviewer's question we add some ideas, how to formulate such a threshold, depending on temperature and vertical velocity. As stated there, this approach might be a starting point towards a more realistic parameterisation.

Please note that the purpose of cirrus parameterizations is to estimate total nucleated ice crystal number concentrations, not to resolve the region around the ice supersaturation maximum. The main differences between homogeneous freezing parameterizations (Barahona & Nenes [2008], Dinh et al. (2016), Kärcher & Lohmann (2002)) may be traced back to how the integration over the supersaturation history is approximated.

We understand the purpose of the nucleation parameterisation as used in coarse grid models. However, the supersaturation is the main control variable for triggering ice nucleation, which in turn produces a certain number concentration from the parameterisation.

Ad VI  I am not sure if this is actually a true statement. My understanding re Koop et al. (2000) has always been that a given choice for j0 is not 'completely arbitrary' but chosen as it relates to a nucleation timescale appropriate for analysing the laboratory experiments. This relevant observation time is roughly given by the inverse of the product of the freezing rate coefficient and the volume of supercooled liquid water in the small droplets investigated. Please check with Koop et al. In Kärcher & Lohmann (2002), we have determined T-dependent freezing thresholds by imposing j0 = 10 for (wet radius) 0.25 µm solution droplets, a typical mean size of freezing droplets in cirrus levels, ignoring the comparatively small dry aerosol core volume.

As we added in the text, there is a (weak) interpretation by using the underlying experimental data of the freezing experiments, and it can be interpreted as a freezing probability as carried out in Kärcher and Lohmann (2002). However, the choice of the value $J = 10^{16} \mathrm{m}^{-3}\mathrm{s}^{-1}$ is still arbitrary but it does not play a major role, as we show in our study.

Ad VII  In the light of the above , I am not aware of any cirrus parameterization scheme that uses a single value for this threshold, at least a temperature dependence is employed. Please check.

It is very clear that we meant a temperature-dependent threshold, but we added this for clarification in the text.

Ad VIII  I would argue that bulk models are not the tools of choice to carry out ice nucleation studies. In the case of homogeneous freezing, early freezing large droplets form a cohort of ice crystals that grows ahead of those forming later at higher supersaturation and on smaller droplets. In this way, a size-dispersed ice crystal spectrum is generated allowing for deposition growth of early formed ice crystals to change (reduce) the supersaturation conditions for later nucleation, as growth rates of µm-crystals are very rapid. It is this self-terminating freezing-relaxation mechanism that eventually determines the total homogeneously nucleated ice numbers, irrespective of updraft speed. Due to the sensitive dependence of the freezing rate coefficient on temperature and water activity (ice supersaturation), it is key to model size-resolved homogeneous freezing, if only approximately analytically, as in our parameterization scheme (Kärcher & Lohmann, 2002). It is also needed to robustly model heterogeneous nucleation events, as ice- nucleating particles activate into ice crystals across a range of supersaturation. The point is, of course, that, by design, bulk models (aka two-moment schemes) are not capable of resolving this competition for available water vapor and therefore do not correctly represent the nucleation pulse and may not robustly predict nucleated ice numbers. Doing so requires a size/mass bin (spectral) approach or particle-based microphysics. For the purpose of studying nucleation in clouds, why not use a detailed model describing all relevant processes properly? Bulk models can only be expected to deliver accurate total homogeneously nucleated icenumbers if they simulate bulk water vapor uptake on freshly nucleated ice crystals accurately. Moreover, modal bulk models based on a fixed functional form of the ice crystal size spectrum (via a constant distribution width such as given in line 717) cannot reproduce the rapid change of the size spread of ice crystals during a freezing event that is ultinmately responsible for the quenching of the supersaturation and the shutting off of the freezing pulse. If one wishes, one might view nucleation as parameterized, i.e., constrained / tuned by assumed parameters in bulk models. Arguably, the "overall good agreement of our simple model with the more sophisticated models" (line 774) might be coincidental.

We respectfully disagree. Bulk models can be used for investigations of scenarios involving ice nucleation, and it could be shown in many studies including parcel models or even LES models, that results from bulk models agree very well with observations. We mention some of these studies in response to reviewer#2. It might be that details of the size spectrum of the underlying aerosol particles cannot be captured by bulk models, but even some changes in the mean size or width of aerosol size distributions can be seen in more complex bulk models (see discussion in Spichtinger and Gierens, 2009). It is merely the fact that the nucleation process itself is very robust in terms of variations that even simpler models can easily represent the main features.
The main question should be formulated, which scales must be represented in a certain detail in order to represent processes in a meaningful way. This question is hard to investigate, but asymptotics might be a key analysis tool for that.

Ad IX  I have a hard time supporting the 'should' in the above summary statement. Rather, the question arises how accurately the bulk model used in the manuscript is able to simulate the degree of overshooting.

Overshooting will always constrain the maximum supersaturation close to (within a few percent) the freezing-relaxation threshold because of the freezing-realaxtion feedback: the higher ice supersaturation overshoots (due to faster cooling), the more droplets freeze, the faster the freezing events terminates. The peak supersaturation attained during a freezing event must be distinguished from the characteristic supersaturation where freezing-relaxation sets in, see discussion in Kärcher et al. [2022].

As we already stated in the manuscript (and we added some text there), it would be more meaningful to compare measurements of supersaturation with the maximum supersaturation one can expect in a nucleation event, than with a threshold which corresponds to a certain value of the nucleation rate and might not be reached because the updraft is too slow. Of course, the fine details of a nucleation event might be seen in a bin/particle model, however, the main features are already represented in bulk models, as discussed above.

Ad X I am not sure why this remark features prominently here. There is no connection made to the topic of the manuscript. Could you better motivation this insertion? Also, Spreitzer et al. (2007) seem to describe a numerical artifact (the occurrence of nucleation cycles in a nucleating air parcel triggered by sedimentation and sustained cooling) that is tied to a coarse spatial resolution (box height) relative to the shallow depth of homogeneous freezing zones. The cycles occur when the timescale of vapor loss due to sedimentation (depending on the layer depth) matches the time scale of supersaturation production that scales in proprtion to the imposed updraft speed. High resolution models (meter resolution in the vertical) show that homogeneous freezing at the top of nucleation layers is a continuous process, see e.g., Lin et al. [2005]

We followed the line of arguments by reviewer#1 for reformulating this part of the manuscript. Essentially, we prescribe the supersaturation in the numerical simulations for a better analysis. We will NOT comment on the study Spreitzer et al. (2017), because this is not the topic of the discussion of the manuscript.

**5 References**

Baumgartner, M., Rolf, C., Grooß, J.-U., Schneider, J., Schorr, T., Möhler, O., Spichtinger, P., and Krämer, M., 2022: New investigations on homogeneous ice nucleation: the effects of water activity and water saturation formulations, Atmospheric Chemistry and Physics, 22, 65-91, https://doi.org/10.5194/acp-22-65-2022

Joos, H., Spichtinger, P., and Lohmann, U., 2009: Orographic cirrus in a future climate, Atmospheric Chemistry and Physics, 9, 7825–7845, https://doi.org/10.5194/acp-9-7825-2009

Kärcher, B. and Lohmann, U., 2002: A parameterization of cirrus cloud formation: Homogeneous freezing of supercooled aerosols, Journal of Geophysical Research: Atmospheres, 107, https://doi.org/10.1029/2001JD000470

Koop, T. and Murray, B. J., 2016: A physically constrained classical description of the homogeneous nucleation of ice in water, Journal of Chemical Physics, 145, https://doi.org/10.1063/1.4962355

Koop, T., Luo, B., Tsias, A., and Peter, T., 2000: Water activity as the determinant for homogeneous ice nucleation in aqueous solutions, Nature, 406, 611-614

Korolev, A. and Mazin, I., 2003: Supersaturation of water vapor in clouds. journal of the atmospheric sciences, 60, 24, 2957-2974.

Krämer, M., Rolf, C., Luebke, A., Afchine, A., Spelten, N., Costa, A., Meyer, J., Zoeger, M., Smith, J., Herman, R. L., Buchholz, B., Ebert, V., Baumgardner, D., Borrmann, S., Klingebiel, M., and Avallone, L., 2016: A microphysics guide to cirrus clouds - Part 1: Cirrus types, Atmospheric Chemistry and Physics, 16, 3463–3483, https://doi.org/10.5194/acp-16-3463-2016

Pruppacher, H. R., 1995: A new look at homogeneous ice nucleation in supercooled water drops, Journal of the Atmospheric Sciences, 52, 1924-1933.

Spichtinger, P., 2014: Shallow cirrus convection - a source for ice supersaturation, Tellus A, 66, 19 937, https://doi.org/10.3402/tellusa.v66.19937

Spichtinger, P. and Gierens, K. M., 2009: Modelling of cirrus clouds - Part 1a: Model description and validation, Atmospheric Chemistry and Physics, 9, 685–706, https://doi.org/10.5194/acp-9-685-2009

Spreitzer, E. J., Marschalik, M. P., and Spichtinger, P., 2017: Subvisible cirrus clouds – a dynamical system approach, Nonlinear Processes in Geophysics, 24, 307-328, https://doi.org/10.5194/npg-24-307-2017

Wernli, H., Boettcher, M., Joos, H., Miltenberger, A. K., and Spichtinger, P., 2016: A trajectory-based classification of ERA-Interim ice clouds in the region of the North Atlantic storm track, Geophysical Research Letters, 43, 6657–6664, https://doi.org/10.1002/2016GL068922

---

## Author Response (AR2)

**Response to comments of reviewer #2**

P. Spichtinger, P. Marschalik, M. Baumgartner,

We thank the reviewer for pointing out a weak formulation.

We list the reviewer's comments (in blue) below together with our responses. We also added a track change version of the respective changes.

Main point:
The authors reasonably argue that the nucleation rate of droplets with activity at water saturation should match that of pure liquid water. What is not clearly explained, in my opinion, is that the original Koop et al., 2000 formula was already derived from this requirement (cf Koop et al., 2000: "Assuming this to be the case also for other values of J, we can compute J(Daw) by fitting to the tabulated J-values of pure water J(a w = 1)". ) Hence, the current inconsistency solely results from the change in the expression of the nucleation rate of pure water. This should be clarified.

We added some text for clarification.

Figure 1: Could you remind the reader which supercooled liquid water saturation vapor pressure is used to compute Jsol . (Though I assume it does not matter too much in that T range.)

We added a sentence about the relevant formulae.

Line 253: (related to the main comment) this statement is misleading, since the original nucleation rate formulation by Koop et al. was constructed to fit the homogeneous nucleation rate of pure water

We have deleted some parts of this statement and added some text for clarification as mentioned above.

---

## Author Response (AR3)

**Betreff:** Re: corrections for acp-2022-434
**Von:** Daniel Knopf <daniel.knopf@stonybrook.edu>
**Datum:** 06.01.23, 17:46
**An:** Peter Spichtinger <spichtin@uni-mainz.de>, ACP Editorial <editorial@copernicus.org>

Dear Prof. Peter Spichtinger,

Indeed, there is significant repetition of context in this paragraph. After careful reviewing and looking at Fig. 2, I am fine with your suggestion to delete the highlighted text sections in the accepted manuscript.

This note and email could be added to your page proofing document to inform the Editorial.

With best regards,

Daniel Knopf

On Fri, Jan 6, 2023 at 7:13 AM Peter Spichtinger <spichtin@uni-mainz.de> wrote:
> Dear Dr Knopf,
>
> I apologize for bothering you again.
>
> During the preparation of the files for the production office I realized that we wrote more or less the same content three times within one paragraph. Therefore I would like to delete some parts of the text. Since this is a change in the already accepted manuscript, I wanted to ask you if this is OK with you. I have attached the relevant page, the marked text would be deleted.
>
> Please let me know if this is OK with you and if (and how) I have to communicate this to the editorial office.
>
> Thank you very much in advance.
>
> Best regards,
> Peter
>
>
>
> --
> --------------------------------------------------------------
> Prof. Dr. Peter Spichtinger
> Theoretical cloud physics
> Institute for Physics of the Atmosphere (IPA)
> Johannes Gutenberg University Mainz
> J.-J.-Becherweg 21, 55128 Mainz, Germany
>
> Office: 05-163
> Phone: +49 (0) 6131 39 - 23157
> Fax:    +49 (0) 6131 39 - 23532

email: spichtin@uni-mainz.de

https://www.staff.uni-mainz.de/spichtin/
https://theoryofclouds.ipa.uni-mainz.de/
https://binary.uni-mainz.de/
https://model.uni-mainz.de/
* * *
--
Daniel A. Knopf (he/him)
Professor of Atmospheric Sciences & Chemistry
School of Marine and Atmospheric Sciences
Department of Chemistry
151 Dana Hall
Stony Brook University
Stony Brook, NY 11794-5000, USA

Tel. (office): +1-631-632-3092
Tel. (lab): +1-631-632-3761
Fax: +1-631-632-6251
E-mail: Daniel.Knopf@stonybrook.edu

More information:
http://you.stonybrook.edu/somas/people/faculty/daniel-knopf/
https://www.stonybrook.edu/commcms/chemistry/faculty/_faculty-profiles/Knopf-Daniel.php
http://you.stonybrook.edu/knopflab/
Researcher ID: F-2040-2011
ORCID: 0000-0001-7732-3922
https://scholar.google.com/citations?user=Qf3D7s0AAAAJ&hl=en&oi=ao

(ii) a Taylor expansion at a prescribed value $y_0$. While the first approach is just a fitting procedure in the relevant range $0.26 \leq \Delta a_w \leq 0.34$, the second approach relies on an a priori choice for the evaluation point $y_0 \in [0.26, 0.34]$ and it is not evident from the outset which value should be used to provide an accurate approximation. For this, we investigate the sensitivity of $p_3$ to a small perturbation $\varepsilon = y - y_0$, i.e. we consider

$$p_3(y) = p_3(y_0 + \varepsilon) = p_3(y_0) + \left.\frac{dp_3}{dx}\right|_{y_0} \varepsilon + \mathcal{O}\left(\varepsilon^2\right) \quad (35)$$

$$\approx b_{t0} + b_{t1} \cdot y = p_{t,y_0}(y) \quad (36)$$

with the coefficients

$$b_{t0} = p_3(y_0) - \left.\frac{dp_3}{dx}\right|_{y_0} \cdot y_0 \quad \text{and} \quad b_{t1} = \left.\frac{dp_3}{dx}\right|_{y_0}. \quad (37)$$

[revised manuscript text omitted]

**4.5 Numerical simulations of nucleation events for different approximations**

In the following we investigate the impact of our approximations of $\log_{10}(J)$ on nucleation events. The setup is as follows: We use the simple bulk ice physics model as described